# Weights to Code: Extracting Interpretable Algorithms from the Discrete Transformer

Yifan Zhang [1 2 †]   Wei Bi [3]   Kechi Zhang [1 2]   Dongming Jin [1 2]   Jie Fu [4 5]   Zhi Jin [1 2]

## Abstract

Algorithm extraction aims to synthesize executable programs directly from models trained on algorithmic tasks, enabling *de novo* recovery of executable mechanisms from weights without relying on human-written target programs. However, applying this paradigm to Transformer is complicated by representation entanglement (e.g., superposition), where features encoded in overlapping directions substantially hinder the recovery of symbolic expressions. We propose the Discrete Transformer, an architecture explicitly designed to bridge the gap between continuous representations and discrete symbolic logic. By injecting discreteness through temperature-annealed sampling, our framework effectively leverages hypothesis testing and symbolic regression to extract human-readable programs. Empirically, the Discrete Transformer achieves performance comparable to the RNN-based MIPS baseline on shared discrete tasks, while broadening extraction to tasks with continuous-valued intermediate computations. Finally, we show that architectural inductive biases provide fine-grained control over synthesized programs, establishing the Discrete Transformer as a controllable testbed for algorithm extraction and Transformer interpretability.

## 1. Introduction

Program synthesis is the task to construct a program that provably satisfies a given high-level formal specification, a long-standing problem which can date back to Church

† Work done during an internship at Kuaishou Technology. [1]Key Laboratory of High Confidence Software Technology (PKU), MOE, Beijing, China [2]School of Computer Science, Peking University, Beijing, China [3]Kuaishou Technology, Beijing, China [4]Shanghai AI Lab, Shanghai, China [5]Shanghai Innovation Institute, Shanghai, China. Correspondence to: Jie Fu <fujie@pjlab.org.cn>, Zhi Jin <zhijin@pku.edu.cn>.

*Proceedings of the 43rd International Conference on Machine Learning*, Seoul, South Korea. PMLR 306, 2026. Copyright 2026 by the author(s).

(1960). In recent years, this field has been revolutionized by Large Language Models (LLMs), which have achieved remarkable success in code generation (Rozière et al., 2024; Guo et al., 2024; CodeGemma Team et al., 2024). Despite their success, an alternative paradigm—*algorithm extraction*—offers a distinct advantage: the ability to recover executable mechanisms *de novo* from trained model weights, without requiring human-written target programs. Recent work has demonstrated the feasibility of this approach in Recurrent Neural Networks (RNNs) via the Mechanistic-Interpretability-based Program Synthesis (MIPS), which successfully leverages symbolic regression to synthesize programs that accurately match neural network behavior (Michaud et al., 2024).

However, extending algorithm extraction to the dominant Transformer architecture faces significant challenges, rooted in a fundamental gap between the continuous, high-dimensional nature of Transformer and the discrete, sparse nature of symbolic algorithms. The primary obstacle lies in interpreting the Transformer's internal representations. Specifically, the standard Transformer often exhibits "superposition", where features are encoded in an overlapping, non-orthogonal set of directions rather than individual neurons (Cunningham et al., 2023; Elhage et al., 2022). Consequently, the model's internal representations can be entangled and polysemantic. Unlike the disentangled input-output mappings required for symbolic regression, these representations make the direct extraction of explicit symbolic expressions challenging. This motivates our central question: *Is it possible to synthesize executable and interpretable programs by extracting algorithms from Transformer?*

In this work, we propose the *Discrete Transformer*, an architecture explicitly optimized for algorithm extraction (see Figure 1). Building upon the framework of Transformer Programs (Friedman et al., 2023), our design incorporates critical modifications to facilitate the transition from continuous dynamics to discrete logic. Architecturally, the Discrete Transformer comprises the Numerical Attention, Numerical MLP, and linear output head. Aligning with the Restricted Access Sequence Processing (RASP) computational model (Weiss et al., 2021), we impose a strict functional disentanglement, where the Numerical Attention is responsible for routing information across positions,

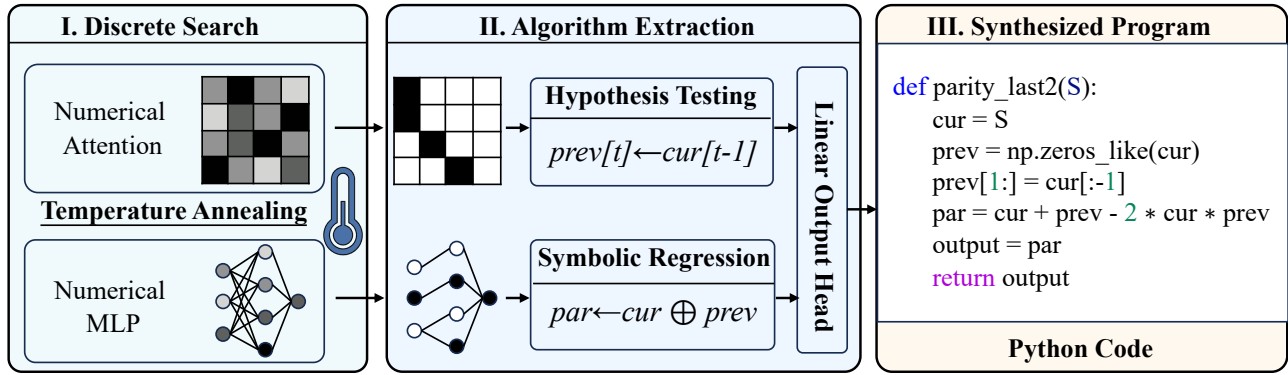

*Figure 1.* Illustration of the proposed framework for extracting executable algorithms from a Discrete Transformer. (I) Discrete Search: Temperature annealing is leveraged to encourage interpretable discretization in both Numerical Attention and MLP modules. (II) Algorithm Extraction: Attention patterns are characterized via hypothesis testing, while arithmetic transformations are approximated through symbolic regression. (III) Synthesized Program: The extracted components are integrated via a linear output head to generate Python code. As shown, the framework successfully recovers the `parity_last2` algorithm, correctly implementing the arithmetic XOR logic.

while the Numerical MLP is dedicated solely to element-wise arithmetic operations. Crucially, we incorporate differentiable sampling mechanisms into each module to inject temperature-controlled discreteness. Through annealing, the model gradually transitions into a fully discrete computation graph, ensuring that the solution to the algorithmic task is implicitly but clearly encoded within its weights.

Once the model converges to a discrete state, we employ a modular strategy to recover the underlying algorithm. Recognizing the distinct roles of the components, we adopt *hypothesis testing* for the Numerical Attention modules to identify interpretable routing patterns, and apply *symbolic regression* to the Numerical MLP modules to infer the specific arithmetic expressions. Finally, these extracted primitives are aggregated through the linear output head, yielding a concise, human-readable, and executable program that verifiably solves the target task.

We validate our approach across a diverse suite of algorithmic tasks, achieving comparable algorithm extraction performance to the RNN-based prior method (MIPS). Crucially, unlike MIPS, which is constrained by the finite-state approximations imposed by post-hoc integer autoencoders, our Discrete Transformer natively processes continuous latent variables, thereby substantially broadening the scope of mechanistic-interpretability-based program synthesis. Beyond performance, we provide empirical analyses of the framework's architectural sensitivity, training dynamics, and controllability. In addition to revealing that functional convergence precedes full structural discretization (Louizos et al., 2018; Savarese et al., 2021), we demonstrate that tailoring architectural constraints imposes strong inductive biases, establishing the Discrete Transformer as a controllable framework for interpretable algorithm discovery.

Overall, our main contributions are as follows:

- We propose the Discrete Transformer, a RASP-aligned architecture that enforces functional disentanglement to bridge continuous optimization and discrete logic.

- We develop a modular extraction pipeline combining hypothesis testing and symbolic regression, extending mechanistic-interpretability-based synthesis to continuous variables.

- We provide empirical analyses of capacity sensitivity, training dynamics, and architectural interventions, showing how inductive biases can steer the extraction toward alternative but equivalent executable programs.

**Conflict of Interest Disclosure.** Y.Z. proposed and developed the Discrete Transformer during an internship at Kuaishou Technology, and W.B. is employed by Kuaishou Technology. The Discrete Transformer is a research model proposed and evaluated in this paper. The authors declare no other financial or substantive conflicts of interest related to this work.

## 2. Related Work

### 2.1. Mechanistic Interpretability in Transformer

The foundation for connecting Transformer with programs is the RASP (Weiss et al., 2021), which abstracts sequence processing into primitives. Building on this abstraction, Tracr (Lindner et al., 2023) compiles such programs *into* Transformer weights, whereas Transformer Programs (Friedman et al., 2023) and Adaptive Transformer Programs (Lai-Dang et al., 2025) tackle the inverse problem of *learning* interpretable structures. By imposing strict architectural

constraints and employing discrete optimization techniques, such as Gumbel-Softmax (Jang et al., 2017), to promote discretization, these approaches produce models that can be deterministically mapped to programs.

While closely related to our work, Transformer Programs and Adaptive Transformer Programs differ significantly in both representation and objective. Specifically, these frameworks translate forward computations into bounded symbolic formalisms, typically constrained to finite grids or categorical variables. In contrast, our approach recovers compact, executable algebraic programs directly from trained weights via symbolic regression. Furthermore, their focus is faithful forward decompilation, while ours is abstract program recovery from learned mechanisms. Consequently, we regard them as related yet conceptually distinct baselines.

### 2.2. Symbolic Regression and Algorithm Extraction

Symbolic regression searches for closed-form expressions balancing accuracy and complexity (Udrescu et al., 2020; Cranmer, 2023). While traditionally used for scientific discovery (Cranmer et al., 2020; Ma et al., 2022) or recently enhanced by LLMs' scientific priors (Shojaee et al., 2025), we leverage it as the core engine for algorithm extraction—synthesizing concise, readable programs directly from trained neural networks.

The innovative MIPS framework (Michaud et al., 2024) pioneers this approach for RNNs. Analogous to the use of Sparse Autoencoders (SAEs) in interpreting features (Cunningham et al., 2023; Bricken et al., 2023), the MIPS employs post-hoc integer autoencoders to approximate continuous latent states into finite states suitable for symbolic regression. In contrast to the MIPS, which relies on auxiliary quantization modules to approximate discreteness, our Discrete Transformer is architecturally designed to learn an interpretable, inherently discrete computation graph, thereby enabling direct and seamless algorithm extraction.

## 3. Discrete Transformer

In this section, we introduce the Discrete Transformer, a specialized architecture purposefully designed for symbolic regression and algorithm extraction. It is structured as a computational graph with clear functional specialization: the Numerical Attention performs explicit variable routing, while the Numerical MLP is responsible for arithmetic computation.

### 3.1. Numerical Residual Stream

The essential difference between the Discrete Transformer and the standard one lies in the structure of the residual stream and the mechanism of information processing. To align with the symbolic nature of algorithm ex-

traction, we define the residual stream not as latent vectors, but as a collection of explicit scalar variables. Let $\mathbf{h}_l = [x_1, \ldots, x_{N_l}]^\top \in \mathbb{R}^{N_l}$ denote the residual stream at layer $l$, consisting of $N_l$ distinct scalar variables.[1] In contrast to standard additive updates, we adapt the concatenation principle (Friedman et al., 2023; Lai-Dang et al., 2025) to the scalar domain, updating the residual stream with new outputs $\mathbf{o}_l$:

$$\mathbf{h}_{l+1} = \text{Concat}(\mathbf{h}_l, \mathbf{o}_l) \in \mathbb{R}^{N_l + |\mathbf{o}_l|}. \tag{1}$$

This design preserves the full computational history as disentangled coordinates, alleviating the information interference induced by superposition in dense vectors.

Moreover, a critical challenge in algorithm extraction is to identify *which* variables serve as operands. Inspired by learnable input selection (Friedman et al., 2023), we address this by designing a discretized reading mechanism tailored for the numerical residual stream: for any computational module (Numerical Attention or MLP) at layer $l$ requiring $k$ inputs, we learn a projection matrix $\mathbf{W}_{\text{read}} \in \mathbb{R}^{k \times N_l}$ to select inputs from the residual stream $\mathbf{h}_l$. To enable discrete structure optimization within a differentiable framework, we apply a temperature-controlled sampling function $S(\cdot, \tau)$ (derivation provided in Appendix A) row-wise to the logits $\mathbf{W}_{\text{read}}$. The input vector $\mathbf{u} \in \mathbb{R}^k$ is then retrieved from the current residual stream $\mathbf{h}_l \in \mathbb{R}^{N_l}$ via:

$$\mathbf{u} = S(\mathbf{W}_{\text{read}}, \tau)\mathbf{h}_l, \tag{2}$$

where $\tau$ is the annealing temperature. As temperature $\tau \to 0$, the selection distribution converges to a deterministic one-hot indicator, effectively functioning as a differentiable pointer over the computation graph.

### 3.2. Numerical Attention as Router

Following the architectural paradigm established in Tracr (Lindner et al., 2023) and Transformer Programs (Friedman et al., 2023), we employ a hard attention mechanism designed strictly as an information routing operator, rather than a feature mixer. In the context of algorithm extraction, this design can help isolate all computational transformations within the Numerical MLP, thereby reducing the difficulty of attention analysis.

**Piecewise Linear Encoding.** Since the residual stream consists of raw scalars ($\mathbf{h}_l \in \mathbb{R}^{N_l}$), distinct scalar variables lack the high-dimensional expressivity required for effective dot-product comparisons. To address this, we define the query and key as selected scalars $x_q, x_k \in \mathbb{R}$, and project

---

[1]Unless otherwise stated, equations in this section and the subsequent extraction procedure are applied position-wise and in parallel over sequence positions; only Numerical Attention operates across positions.

them into a higher-dimensional space using the Piecewise Linear Encoding (PLE) (Gorishniy et al., 2022):

$$\mathbf{q} = \phi_{\text{PLE}}(x_q), \quad \mathbf{k} = \phi_{\text{PLE}}(x_k), \tag{3}$$

where $\mathbf{q}, \mathbf{k} \in \mathbb{R}^{d_{\text{attn}}}$ and $\phi_{\text{PLE}} : \mathbb{R} \to \mathbb{R}^{d_{\text{attn}}}$ is a learnable mapping.

**Deterministic Attention Mechanism.** The attention scores $\mathbf{a}_{\text{raw}} \in \mathbb{R}^T$ over a context length $T$ are computed via scaled dot-product with T5-style relative positional biases $\mathbf{b}_{\text{rel}}$ (Raffel et al., 2023):

$$\mathbf{a}_{\text{raw}} = \frac{\mathbf{K}\mathbf{q}}{\sqrt{d_{\text{attn}}}} + \mathbf{b}_{\text{rel}}, \tag{4}$$

where $\mathbf{K} \in \mathbb{R}^{T \times d_{\text{attn}}}$ stacks the projected keys from the context. To ensure interpretable attention patterns, we enforce hard attention using the sampling function $S(\cdot, \tau)$. Crucially, the value projection is the identity function for scalars, meaning that the value vector $\mathbf{v} \in \mathbb{R}^T$ consists directly of the raw scalars from the history. The output $o_{\text{attn}} \in \mathbb{R}$ is a weighted sum:

$$o_{\text{attn}} = S(\mathbf{a}_{\text{raw}}, \tau)^\top \mathbf{v}. \tag{5}$$

This design forces the attention head to converge to a deterministic pointer operation, retrieving specific raw values from history (e.g., "copy the value from the token at offset -1").

### 3.3. Numerical MLP as Operator

While the Numerical Attention handles data movement, the Numerical MLP is dedicated to element-wise arithmetic and logical transformations. Drawing on the insight that Transformer MLPs contribute additive updates to the residual stream, which can be decomposed into weighted sums of sub-updates (Geva et al., 2022), we explicitly decompose the MLP module into a collection of parallel, independent sub-modules (sub-MLPs).

Each sub-module performs a single elementary operation. It first selects a small, fixed number of scalars (typically $k = 2^2$) from the stream via the discretized reading mechanism, forming an operand vector $\mathbf{u} \in \mathbb{R}^k$. These operands are processed by a shallow network:

$$o_{\text{mlp}} = \mathbf{W}_2 \sigma(\mathbf{W}_1 \mathbf{u} + \mathbf{b}_1) + b_2, \tag{6}$$

where $\mathbf{W}_1 \in \mathbb{R}^{d_{\text{hid}} \times k}$, $\mathbf{W}_2 \in \mathbb{R}^{1 \times d_{\text{hid}}}$ are learnable weights, $\mathbf{b}_1, b_2$ are biases, and $\sigma$ is a non-linear activation (e.g., ReLU). The resulting scalar output $o_{\text{mlp}}$ is concatenated back to the residual stream.

---

[2]Unless otherwise specified, we set $k = 2$ to capture binary interactions while keeping symbolic regression low-dimensional and stable; see Section B.

**Inductive Bias for Symbolic Regression.** By severely constraining the input dimension $k$ and the hidden width $d_{\text{hid}}$, we deliberately limit the complexity of each sub-module. This bottleneck forces the network to decompose complex functions into a sequence of simple arithmetic steps, creating ideal conditions for symbolic regression (e.g., PySR (Cranmer, 2023)) to extract closed-form expressions.

### 3.4. Linear Output Head

The architecture concludes with a linear output head that aggregates the discrete computational steps into a final prediction:

$$\hat{y} = \mathbf{w}_{\text{out}}^\top \mathbf{h}_{\text{final}}, \tag{7}$$

where $\mathbf{h}_{\text{final}}$ is the final residual stream, and $\mathbf{w}_{\text{out}}$ represents the aggregation weights. The final trained model thus represents a clean, executable computational graph: nodes correspond to either routing operations (Numerical Attention) or arithmetic functions (Numerical MLP), and edges are defined by the discrete selections of the discretized reading modules.

## 4. From Weights to Code

After training with temperature annealing, the converged Discrete Transformer reaches a fully discrete state, and the algorithmic solution is implicitly encoded within its sparse weights and activation patterns. To recover an explicit, human-readable program, we treat the trained model as a computational graph composed of two distinct types of nodes: *routers* (Attention) and *operators* (MLP). We propose a decoupled extraction pipeline that first infers the function of each node using component-specific strategies, then reconstructs the global program trace via backward traversal.

### 4.1. Hypothesis Testing for Numerical Attention

For the Numerical Attention module, directly extracting explicit symbolic expressions via symbolic regression is challenging due to the computational complexity introduced by the PLE of queries and keys, dot-product interactions, and the hard attention mechanism induced by sampling functions. However, since the structural role of attention is constrained to be an information router, we abstract away from the intermediate arithmetic operations and focus our interpretability analysis on the resulting routing patterns.

We conceptualize the Numerical Attention module as a deterministic pointer performing differentiable addressing on context memory. Following Neural Turing Machines (Graves et al., 2014), we categorize addressing into *Location-based* and *Content-based*. We hypothesize that our attention heads specialize into these two modes, and empirically observe that they manifest as two typical interpretable patterns:

*Fixed Offset* and *Windowed Extrema*.[3]

Specifically, for a given head, we analyze its attention matrices generated over the validation set and test the hypotheses by examining the statistical properties:

- *Fixed Offset.* This pattern corresponds to relative positional indexing. We hypothesize that there exists an integer offset $\delta$ such that, for a large fraction of query positions $i$, the head places most of its attention on $j = i - \delta$. We operationalize this by measuring whether the averaged attention mass concentrates on a single offset diagonal (corresponding to $j = i - \delta$) beyond a predefined threshold.

- *Windowed Extrema.* This pattern reflects content-dependent selection. Let $\mathbf{v} \in \mathbb{R}^T$ be the sequence of scalar values, where $v_j$ is the value at position $j$. We hypothesize that there exists a window size $z \in \mathbb{Z}^+$ such that, for a large fraction of query positions $i$, the head primarily attends to $j^* = \arg\min/\max_{j \in \{i-z+1,\dots,i\}} v_j$. We verify this hypothesis by checking the sample-wise agreement between the head's selected index and the true extremum index within the window.

There might exist "unmatched heads"; however, across the algorithm benchmark, we find that they represent computational noise—being negligible in magnitude or acting as redundant variables—and do not influence the logic of the final assembled program (see Appendix C for detailed analysis).

### 4.2. Symbolic Regression for Numerical MLP

In contrast to the Numerical Attention, the Numerical MLP modules serve as the arithmetic core. Thanks to the disentangled architecture, each MLP sub-module functions as an isolated mapping $f : \mathbb{R}^k \to \mathbb{R}$ with low-dimensional inputs. This architectural isolation makes them ideal candidates for black-box symbolic regression.

For each sub-MLP, we collect a dataset of input-output pairs $\mathcal{D} = \{(\mathbf{u}^{(i)}, o_{\text{mlp}}^{(i)})\}_{i=1}^M$ from validation passes. We employ the PySR (Cranmer, 2023), a symbolic regression tool based on genetic algorithms, to search for the optimal symbolic expression $\hat{f}$ that minimizes the error on $\mathcal{D}$. Crucially, the constrained input dimension and limited model capacity drastically reduce the search space, enabling the PySR to reliably converge to exact arithmetic expressions (e.g., $o_{\text{mlp}} = 2u_1 + u_2$) rather than approximate fits.

### 4.3. Global Program Assembly

The final phase integrates these extracted primitives into a coherent program. The linear output head, $\hat{y} = \mathbf{w}_{\text{out}}^\top \mathbf{h}_{\text{final}}$, serves as the entry point for extraction. To reduce the length of the extracted program, we impose a sparsity threshold $\epsilon$ on the magnitude of the weights $\mathbf{w}_{\text{out}}$. This prunes unnecessary variables from the computation graph, retaining the active variables contributing to the final prediction.

Starting from these active variables, we perform a *backward traversal* of the computational graph. Recursively, we replace each intermediate variable in the residual stream with its corresponding symbolic definition, either a deterministic pointer from the Numerical Attention or a distilled arithmetic expression from the Numerical MLP, until the traversal reaches the raw input tokens. This process effectively compiles the neural network into concise, closed-form algorithmic expressions that approximate the Discrete Transformer's behavior with high fidelity across the relevant input domain.

## 5. Experiments

In this section, we evaluate the Discrete Transformer on a diverse suite of algorithmic reasoning tasks. We aim to demonstrate that, beyond achieving high predictive performance, our model extracts interpretable and human-readable algorithms with high fidelity, thereby revealing the underlying structure and logic inherent in the data.

**Datasets** We construct an algorithmic reasoning benchmark designed to evaluate symbolic rule recovery across a range of capabilities, including arithmetic reasoning, variable tracking, and non-linear composition. The tasks are primarily sourced from the MIPS benchmark (Michaud et al., 2024), excluding those whose ground-truth rules cannot be expressed using general symbolic operators (e.g., the palindrome rule in `bit_palindrome`).[4] To broaden coverage of continuous-valued physical dynamics, we include five continuous-valued tasks: `exponential_moving_average`, `free_fall_height`, `linear_drop`, `quadratic_drop`, and `damped_harmonic_oscillator`. We group all tasks into three categories based on their underlying logic: (1) *Linear Arithmetic* (e.g., `sum_last2`, `gravity`), which involves learning linear mathematical or physical logic; (2) *Non-Linear Composition* (e.g., `parity_last2`, `add_mod_3`), which requires approximating non-linear logic; and (3) *Continuous Dynamics* (e.g., `exponential_moving_average`), which assess the capability to model tasks with floating-

---

[3]These patterns are not intended as a complete taxonomy of attention behaviors; see Appendix D for further discussion.

[4]Since our goal is to extract closed-form expressions of fixed-size through symbolic regression (typically PySR), we restrict our benchmark to tasks whose ground-truth mappings are representable under PySR's default operator set (e.g., $+, -, \times, \div$).

point variables. Detailed task definitions are provided in Appendix Table 5.

**Training Details** All models adopt a decoder-only architecture implemented in PyTorch, optimized via AdamW to minimize the mean squared error (MSE). We train for 50 epochs with a batch size of 512 and cosine learning rate decay. Crucially, to handle discrete optimization, we apply geometric annealing to the sampling temperature $\tau$, decreasing it from 10.0 to 0.1. We report the average performance across three random seeds, with hyperparameters (layers, heads, sub-MLPs) selected via grid search. Full experimental details are provided in Appendix F.

**Results** The Discrete Transformer demonstrates strong predictive performance across most of the algorithmic tasks, with MSE approaching zero (see Appendix Table 5). Beyond prediction, our primary contribution is to show that the solutions encoded in the Discrete Transformer can be *explicitly* extracted into human-readable code. Specifically, by applying the methodology described in Section 4, we successfully distill executable Python programs from the trained weights. As shown in Table 1, the extracted programs successfully solve most algorithmic tasks, achieving an accuracy of 1.00, which matches the performance of MIPS. Moreover, the low root mean squared error (RMSE) further indicates the high fidelity of our extraction methodology. Notably, the Discrete Transformer additionally handles continuous-dynamics tasks that lie beyond the scope of MIPS. We summarize three key findings below.

- **Discovery of Algorithmic Mechanisms in Linear Tasks.** For linear tasks, the discrete optimization process exhibits an efficient ability to discover underlying algorithmic mechanisms. On simple instances, the model frequently learns to bypass the Numerical MLP entirely, instead leveraging the linear output head to perform arithmetic directly. For instance, in `sum_last2` (Figure 2),[5] the model assigns specific attention heads to retrieve $x_{t-1}$ (via a verified "Fixed Offset" pattern) and integrates it with the current token $x_t$. Symbolic simplification yields the exact expression $y_t = x_t + x_{t-1}$,[6] effectively capturing the underlying addition logic. On more complex physical tasks such as `gravity`, the model identifies the essential computational variables and constructs the correct computation graph through the coordination of the Numerical Attention, Numerical MLP, and linear output head, demonstrating strong potential for efficient modeling of physical processes.

- **Exact Recovery of Non-Linear Identities.** For tasks requiring non-linear logic, the extraction methodology successfully identifies algebraically equivalent expressions rather than merely producing approximate fits. For instance, in the `parity_last2` task ($x_t \oplus x_{t-1}$) with binary inputs $x \in \{0, 1\}$, simplifying the extracted expressions (Figure 3) yields $y_t = x_t + x_{t-1} - 2x_t x_{t-1}$, which is an exact algebraic formulation of the parity operation. For `maximum_prev2` and `minimum_prev2` tasks, simplifying the extracted expressions in Figure 4 reveals a piecewise-linear implementation of conditional computation via ReLU operations: $\max(x_t, x_{t-1}) = x_{t-1} + \text{ReLU}(x_t - x_{t-1})$ and $\min(x_t, x_{t-1}) = x_{t-1} - \text{ReLU}(x_{t-1} - x_t)$. This demonstrates the capability to reverse-engineer blackbox neural computations into explicit algebraic forms equivalent to logical operations.

- **Modeling Continuous Dynamics.** Owing to its inherent model design, our framework offers a distinct qualitative advantage over prior symbolic synthesis approaches in terms of broader applicability. MIPS, in particular, relies on discrete latent-state abstractions and is therefore not directly applicable to floating-point data types. In contrast, the Discrete Transformer natively operates on continuous variables, allowing the extracted programs to preserve real-valued intermediate computations rather than forcing discretization. This distinction enables the extraction of algorithms that capture continuous dynamics in tasks such as `exponential_moving_average` and `free_fall_height`, substantially broadening the scope of mechanistic-interpretability-based program synthesis.

To evaluate the robustness of our framework under architectural variations, we examine how key hyperparameters, including the number of layers, attention heads, and sub-MLPs, affect both convergence (MSE) and program complexity (measured by the line count). Across the tested tasks and architectural configurations, our results indicate that once the architecture meets the minimal functional requirements of the task, the Discrete Transformer generally attains near-zero MSE loss and yields executable extracted programs.

Over-parameterized architectures remain robust in terms of task performance, but they can lead to increased redundancy in the extracted programs. In particular, multiple heads or sub-MLPs may learn functionally similar operations, increasing the program line count without yielding commensurate gains in accuracy. We find that such redundancy can be substantially reduced by imposing a mild simplicity bias, such as $\ell_1$ regularization on the linear output head, while maintaining low extraction error. Moreover, excess

---

[5]For clarity of presentation, we round the coefficients of symbolic expressions to two decimal places.

[6]We employ the SymPy library (Meurer et al., 2017) to automatically simplify the extracted expressions, leading to more concise and readable mathematical formulae.

*Table 1.* Comparison of algorithm extraction performance between MIPS and the Discrete Transformer (Ours) on the algorithmic benchmark. For integer-valued tasks, we evaluate test-set accuracy (Acc.) by rounding outputs to the nearest integer and report Root Mean Squared Error (RMSE) to assess extraction fidelity. For tasks involving floating-point data types, Acc. is not applicable (N/A) and only RMSE is reported. MIPS results are retrieved from Michaud et al. (2024); † denotes that MIPS fails to achieve perfect accuracy (1.00) or is inapplicable due to inherent limitations.

| Task | MIPS | Ours | | Task | MIPS | Ours | |
|---|---|---|---|---|---|---|---|
| | Acc. | Acc. | RMSE | | Acc. | Acc. | RMSE |
| *Linear Arithmetic* | | | | | | | |
| `sum_last2` | 1.00 | 1.00 | $8.22 \times 10^{-5}$ | `diff_last2` | 1.00 | 1.00 | $2.65 \times 10^{-8}$ |
| `sum` | 1.00 | 1.00 | $1.30 \times 10^{-7}$ | `div_3` | † | 0.67 | $4.69 \times 10^{-1}$ |
| `freebody` | 1.00 | 1.00 | $9.78 \times 10^{-6}$ | `gravity` | 1.00 | 1.00 | $4.31 \times 10^{-6}$ |
| `spring` | 1.00 | 1.00 | $5.91 \times 10^{-5}$ | `magnetic` | † | 0.02 | $1.06 \times 10^{1}$ |
| *Non-Linear Composition* | | | | | | | |
| `parity_last2` | 1.00 | 1.00 | $1.21 \times 10^{-6}$ | `maximum_prev2` | 1.00 | 1.00 | $2.10 \times 10^{-3}$ |
| `minimum_prev2` | 1.00 | 1.00 | $4.21 \times 10^{-3}$ | `unique_prev2` | 1.00 | 1.00 | $2.05 \times 10^{-7}$ |
| `bitwise_and` | 1.00 | 1.00 | $1.03 \times 10^{-9}$ | `bitwise_or` | 1.00 | 1.00 | $1.28 \times 10^{-3}$ |
| `bitwise_not` | 1.00 | 1.00 | $3.80 \times 10^{-9}$ | `bitwise_xor` | 1.00 | 1.00 | $2.58 \times 10^{-9}$ |
| `balanced_parenthesis` | † | 0.98 | $1.45 \times 10^{-1}$ | `abs` | 1.00 | 1.00 | $2.29 \times 10^{-7}$ |
| `abs_of_diff` | 1.00 | 1.00 | $1.42 \times 10^{-4}$ | `parity` | 1.00 | 1.00 | $8.99 \times 10^{-9}$ |
| `parity_zeros` | 1.00 | 1.00 | $1.99 \times 10^{-5}$ | `add_mod_3` | 1.00 | 1.00 | $4.15 \times 10^{-4}$ |
| `bit_dot_prod_mod_2` | 1.00 | 1.00 | $1.97 \times 10^{-3}$ | `bit_addition` | 1.00 | 0.99 | $1.49 \times 10^{-1}$ |
| *Continuous Dynamics* | | | | | | | |
| `exponential_moving_average` | † | N/A | $4.04 \times 10^{-7}$ | `free_fall_height` | † | N/A | $9.02 \times 10^{-3}$ |
| `linear_drop` | † | N/A | $5.21 \times 10^{-4}$ | `quadratic_drop` | † | N/A | $9.37 \times 10^{-4}$ |
| `damped_harmonic_oscillator` | † | N/A | $3.18 \times 10^{-4}$ | | | | |

capacity may enlarge the discrete search space and make optimization less stable, but a slower temperature annealing schedule alleviates this effect. Overall, these results indicate that the Discrete Transformer is robust across a range of architectural choices: once sufficient functional capacity is available, it reliably recovers accurate discrete solutions, while the additional complexity introduced by excess capacity can be controlled through regularization and annealing. Additional details are provided in Appendix G.

# 6. Further Analysis

In this section, we provide empirical analyses of the Discrete Transformer's properties, positioning it at the intersection of program synthesis and continuous sparsification. We primarily focus on the training dynamics, characterizing a distinct phase where functional convergence is achieved prior to complete structural discretization. We further demonstrate how architectural constraints serve as strong inductive biases, establishing the model as a controllable testbed for interpretable algorithm discovery.

## 6.1. Continuous-to-Discrete Homotopy

We frame program synthesis as a continuous-to-discrete homotopic transformation. From the perspective of differentiable architecture search and continuous sparsification (Louizos et al., 2018; Jang et al., 2017), we relax the discrete

constraint by introducing continuous structural parameters, specifically the projection matrices $\mathbf{W}_{\text{read}}$. The temperature $\tau$ serves as a homotopy parameter: high values define a search space over the continuous probability simplex for exploration, while annealing $\tau \to 0$ smoothly deforms the distribution toward simplex vertices for exploitation.

To validate these dynamics, in addition to the MSE loss (serving as the soft training objective $\mathcal{L}_{\text{soft}}$), we monitor two metrics: Structural Agreement ($\mathcal{A}(e)$) and Discretization Discrepancy ($\Delta(e)$). Let $\{\mathbf{p}_r^{(e)}\}_{r=1}^R$ be the set of all row-wise probability distributions derived from $S(\mathbf{W}_{\text{read}}, \tau)$ at epoch $e$, $E$ the total number of epochs, and $\mathcal{L}_{\text{hard}}(e)$ the hard loss computed via deterministic argmax sampling. We define $\mathcal{A}(e) = \frac{1}{R} \sum_{r=1}^R \mathbb{I}\left(\arg\max \mathbf{p}_r^{(e)} = \arg\max \mathbf{p}_r^{(E)}\right)$. And we further define $\Delta(e) = \mathcal{L}_{\text{hard}}(e) - \mathcal{L}_{\text{soft}}(e)$. As shown in Figure 5, we observe a distinct phase transition: the significant decline in $\Delta(e)$ occurs slightly later than that of $\mathcal{L}_{\text{soft}}$, concurrently with $\mathcal{A}(e)$ approaching 1.0 as annealing proceeds. This lag suggests a two-stage process where the model first achieves *soft convergence*—learning functional mappings via relaxed representations—before undergoing *structural crystallization*, thereby ensuring a robust transition from exploration to exploitation.

```python
import numpy as np
def sum_last2(input_seq):
    # V0_input
    input_arr = np.array(input_seq, dtype
    =float)
    seq_len = input_arr.shape[0]
    V0 = input_arr[:, 0]
    # V1_Attn_L0H0
    V1 = np.zeros(seq_len)
    # Fixed offset 1
    V1[1:] = V0[:-1]
    # V2_Attn_L0H1
    V2 = np.zeros(seq_len)
    # Fixed offset 1
    V2[1:] = V0[:-1]
    # Output Head
    output = 1.00 * V0 + 1.00 * V1 + 0.01
    * V2
    return output
```

⇓ *symbolic simplification*

**Simplified Expression:** $\quad y_t = x_t + x_{t-1}$

*Figure 2.* Algorithm extraction results for the `sum_last2` task. Modules are denoted by their type and indices (e.g., `Attn_L0H0` represents the attention head at index 0 of layer 0). The extracted code reveals that the model utilizes specific attention heads to retrieve the previous token. Symbolic simplification (bottom) shows the mathematically simplified expression, verifying that the model correctly learns the logic $y_t = x_t + x_{t-1}$.

```python
import numpy as np
def parity_last2(input_seq):
    # V0_input
    input_arr = np.array(input_seq, dtype
    =float)
    seq_len = input_arr.shape[0]
    V0 = input_arr[:, 0]
    # V1_Attn_L0H0
    V1 = np.zeros(seq_len)
    # Fixed offset 1
    V1[1:] = V0[:-1]
    # V2_Attn_L0H1
    V2 = np.zeros(seq_len)
    # Fixed offset 1
    V2[1:] = V0[:-1]
    # V3_MLP_L0M0
    V3 = ((V1 + -0.41) * ((V0 - 0.41) *
    3.57)) + -0.61
    # Output Head
    output = -0.56 * V3 + 0.17 * V0 +
    0.10 * V2 + 0.07 * V1
    return output
```

⇓ *symbolic simplification*

**Simplified Expression:** $\quad y_t = x_t + x_{t-1} - 2x_t x_{t-1}$

*Figure 3.* Algorithm extraction results for the `parity_last2` task. The extracted code reveals that the model utilizes specific attention heads (e.g., `Attn_L0H0`) to retrieve the previous token, and specific sub-MLPs (e.g., `MLP_L0M0`) to perform non-linear transformations. The bottom panel presents the symbolic simplification $y_t = x_t + x_{t-1} - 2x_t x_{t-1}$, which is the algebraic formulation of the parity logic.

### 6.2. Controllability via Inductive Biases

Unlike standard code LLMs, which synthesize programs based on opaque statistical patterns, our Discrete Transformer offers a unique advantage: interpretability-aware controllability. In scientific discovery, researchers often seek not just any solution, but a specific form of algorithm that aligns with domain knowledge (Schmidt & Lipson, 2009; Udrescu & Tegmark, 2020; Cranmer et al., 2020). We address this need by explicitly manipulating the architectural constraints and training configurations of our model to impose inductive biases, thereby steering the solution space.

We demonstrate this controllability through two intervention scenarios. First, we manipulate architectural primitives. In the `maximum_prev2` task, the unconstrained model typically solves maximization via MLP-based arithmetic approximation (exploiting ReLU non-linearity). To test if the model can switch algorithmic paradigms, we explicitly set the number of sub-MLPs to zero, removing its capacity for complex arithmetic. Consequently, the model adapts by shifting its entire mechanism to the Numerical Attention module. It discovers a "windowed max" attention pattern, solving the task by directly copying the largest value from the context rather than computing it. Second, we intervene in the information flow. In the `spring`

task, the model typically learns the standard recurrence $y_t = y_{t-1} - y_{t-2} + x_t$. By masking the immediate history $(y_{t-1}, y_{t-2})$, we guide the model to bypass the standard path and discover a mathematically equivalent high-order recurrence: $y_t = -y_{t-3} + x_t + x_{t-1}$. These findings highlight the Discrete Transformer as a robust tool for intervenable algorithm discovery, capable of uncovering multiple equivalent logical paths underlying the same data distribution.

## 7. Limitations

The Discrete Transformer intentionally trades expressivity for interpretability. Its router-style attention is designed mainly for explicit information routing, and its sub-MLPs operate with bounded fan-in. Consequently, an $L$-layer model can be viewed as a selector-augmented bounded-depth computation graph, where information from many positions is aggregated progressively across layers. Computations requiring broad aggregation or state propagation may therefore require depth that grows with the sequence length. Our experiments are conducted at benchmark scale with relatively small models, and are intended to establish a controlled framework for high-fidelity program extraction.

```
1  import numpy as np
2  def relu(x):
3      return np.maximum(0, x)
4  def extrema_prev2(input_seq, mode):
5      # V0_input
6      input_arr = np.array(input_seq, dtype
       =float)
7      seq_len = input_arr.shape[0]
8      V0 = input_arr[:, 0]
9      # V1_Attn_L0H0
10     V1 = np.zeros(seq_len)
11     # Fixed offset 1
12     V1[1:] = V0[:-1]
13     # V3_MLP_L0M0, Output Head
14     if mode == "max":  # maximum_prev2
15         V3 = (((V0 * -0.12) + relu((V1 *
           -1.00) + V0)) * -2.21) + ((V1 *
           -0.26) / 1.02)
16         output = 0.88 * V1 + -0.45 * V3 +
            0.12 * V0
17     else:  # minimum_prev2
18         V3 = 1.43 * (V0 + (((3.49 * relu(
           V1 - V0)) - V1) - (V1 * 1.04)))
19         output = 0.42 * V1 + 0.29 * V0 +
           -0.20 * V3
20     return output
```

$$\Downarrow \text{ symbolic simplification}$$

**Simplified Expressions:**

$$\max(x_t, x_{t-1}) = x_{t-1} + \text{ReLU}(x_t - x_{t-1})$$
$$\min(x_t, x_{t-1}) = x_{t-1} - \text{ReLU}(x_{t-1} - x_t)$$

*Figure 4.* Algorithm extraction results for `maximum_prev2` and `minimum_prev2` tasks. The top panel shows the raw code where sub-MLPs utilize ReLU functions to compare the current token $x_t$ with the previous token $x_{t-1}$. The bottom panel presents the simplified expressions, verifying that the model correctly reconstructs the extrema functions using the ReLU-based algebraic identities.

Extending this approach to larger models, richer program classes, and more compositionally complex tasks remains an important direction for future work.

## 8. Conclusion

In this work, we present the Discrete Transformer, a novel architecture enabling program synthesis via algorithm extraction directly from Transformer architectures. By combining a disentangled numerical residual stream with a smooth discrete optimization curriculum, the model separates information routing from arithmetic computation and converges to a discrete, extractable representation. We then recover the underlying algorithm through a modular extraction procedure, using hypothesis testing to identify interpretable routing patterns and symbolic regression to derive the corresponding arithmetic expressions. Across a range of algorithmic benchmarks, our experiments show that this framework achieves

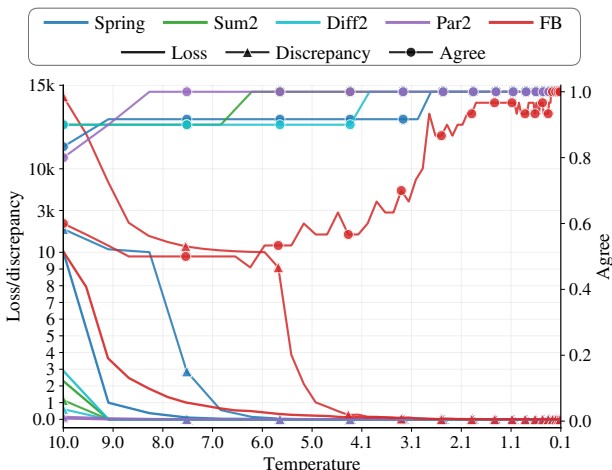

*Figure 5.* Training dynamics exhibit a distinct phase transition: the loss decreases earlier, while the pronounced drop in Discrepancy occurs slightly later, particularly for more challenging tasks such as `spring` and `freebody`, coinciding with Agreement approaching 1.0 during temperature annealing from 10.0 to 0.1. The abbreviations Spring, Sum2, Diff2, Par2, and FB denote the `spring`, `sum_last2`, `diff_last2`, `parity_last2`, and `freebody` tasks, respectively.

performance comparable to RNN-based approaches, while offering fine-grained controllability and significantly broadening the applicability of mechanistic-interpretability-based synthesis.

More broadly, our results suggest that architectural constraints can make Transformer computation more transparent while preserving the ability to solve nontrivial algorithmic tasks. Rather than treating interpretability as a post-hoc analysis of opaque models, the Discrete Transformer embeds interpretability directly into both the architectural design and optimization process. We hope this work provides a step toward faithful algorithm extraction from neural models and motivates future research on bridging interpretability with richer compositional expressivity.

## Acknowledgments

This research is supported by the National Natural Science Foundation of China under Grant Nos. 62192731.

## Impact Statement

This paper advances the fields of mechanistic interpretability and algorithm extraction by enabling the discovery of verifiable, human-readable programs from the Discrete Transformer. There are many potential societal consequences of our work, none of which we feel must be specifically highlighted here.

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

# A. Smooth Transition Mechanism for Discrete Optimization

To train the discrete selection parameters in our discretized reading and Numerical Attention modules, we address the non-differentiability of hard selection while avoiding the pitfalls of standard relaxation methods. While Gumbel-Softmax enables differentiable sampling, it often struggles to escape local optima and fails to promote the strict sparsity essential for interpretability. To address this, we incorporate a smooth transition mechanism (Lai-Dang et al., 2025), which dynamically interpolates between Gumbel-Softmax and Sparsemax (Martins & Astudillo, 2016) throughout the training process. The hybrid sampling vector $\mathbf{p} \in \mathbb{R}^{N_l}$ is defined as:

$$\mathbf{p} = (1 - \alpha(\tau))\mathbf{p}_{\text{soft}} + \alpha(\tau)\mathbf{p}_{\text{sparse}}, \tag{8}$$

where $\mathbf{p}_{\text{soft}}$ and $\mathbf{p}_{\text{sparse}}$ denote sample vectors drawn from Gumbel-Softmax and Gumbel-Sparsemax distributions, respectively. The interpolation factor $\alpha(\tau)$ is a temperature-dependent function, defined as $\alpha(\tau) = \frac{\tau_1 - \tau}{\tau_1 - \tau_2}$ (where $\tau_1$ denotes the initial temperature and $\tau_2$ denotes the final temperature), which monotonically transitions from 0 to 1 as the temperature $\tau$ is geometrically annealed. This mechanism allows the model to prioritize exploration via Softmax in the early stages, before smoothly transitioning to Sparsemax to enforce sparsity and deterministic selection (exploitation) in later stages. Ultimately, this ensures convergence to concise, interpretable program structures.

# B. Arity Ablation

We use $k = 2$ by default because it is the smallest arity that captures many binary arithmetic primitives while keeping symbolic regression tractable.

Ablations show the expected trade-off. For a small model with Layers $= 2$, Heads $= 2$, and sub-MLPs $= 2$, $k = 1$ underfits interaction-heavy tasks such as `add_mod_3` and `bit_dot_prod_mod_2`, yielding RMSE $6.46 \times 10^{-2}$ and $2.36 \times 10^{-1}$, whereas $k = 2$ reduces them to $3.10 \times 10^{-4}$ and $3.00 \times 10^{-4}$. Increasing arity further does not uniformly help: $k = 4$ gives $2.98 \times 10^{-3}$ and $8.44 \times 10^{-4}$ on the same tasks, indicating a larger symbolic regression search space. For `parity_last3`, which requires a genuine three-way interaction, $k = 3$ performs best with RMSE $3.33 \times 10^{-6}$, while $k = 2$ fails ($4.69 \times 10^{-1}$) and $k = 4$ also succeeds but less cleanly ($2.06 \times 10^{-3}$). The same trend appears in a larger model ($L = 4, H = 4, M = 4$): increasing $k$ from 2 to 4 worsens `add_mod_3` and `bit_dot_prod_mod_2`, while only modestly improving `parity_last3`.

# C. Unmatched Heads

Our analysis reveals that the "unmatched heads" do not represent missing algorithmic primitives, but rather manifest as computational noise or redundancy. Specifically, we observe that these heads are generally rendered ineffective through two primary mechanisms: by being suppressed with negligible magnitudes, or by being ignored by higher-layer modules.

For instance, in the `bitwise_and` task, the head labeled `Attn_L0H1` exhibits a disordered attention matrix that matches neither location-based nor content-based patterns. However, symbolic tracing of the computational graph confirms that its output variable is an orphan variable: it is not selected as an input by any subsequent modules nor aggregated by the final output projection. Similarly, in the `minimum_prev2` task with the number of sub-MLPs constrained to zero, we observe that the head labeled `Attn_L0H1` behaves as an unmatched head. Since its output is disconnected from the valid execution path, this lack of interpretability does not impede successful extraction of the underlying algorithm.

# D. Attention Patterns

The attention patterns considered in Section 4.1, namely *Fixed Offset* and *Windowed Extrema*, are not intended to constitute a complete taxonomy of attention behaviors. Instead, they represent the dominant and typically interpretable head classes induced by our architecture and benchmark suite.

For tasks in which attention naturally performs computations beyond pure routing, the set of hypothesized attention classes can be extended without changing the overall extraction pipeline. As diagnostic examples, we consider two additional tasks: `retrieve_value_by_key` and `prefix_mean`. In `retrieve_value_by_key`, we introduce an equality-retrieval head class, which selects values according to key equality rather than relative position or local extrema. In `prefix_mean`, we retain soft attention and regard mean aggregation as an additional computational operator. These extensions preserve the structure of the backward reconstruction procedure, yielding extracted programs with RMSEs of $1.28 \times 10^{-10}$ on

*Table 2.* Effect of output-head pruning on extraction fidelity and program length. Values in parentheses are without pruning.

| Task | RMSE | Program length |
|------|------|----------------|
| `sum_last2` | $1.05 \times 10^{-4}$ ($1.05 \times 10^{-4}$) | 28 (34) |
| `diff_last2` | $0$ ($2.00 \times 10^{-8}$) | 22 (33) |
| `parity_last2` | $1.49 \times 10^{-8}$ ($1.49 \times 10^{-8}$) | 34 (34) |

`retrieve_value_by_key` and $3.14 \times 10^{-4}$ on `prefix_mean`, respectively, thereby demonstrating that the framework can accommodate additional interpretable attention mechanisms beyond the two default patterns.

## E. Magnitude-based Pruning

To reduce the length of the extracted program, we apply a magnitude threshold $\epsilon$ to the output weights. This is a heuristic pruning step: it removes variables that are not needed by the reconstructed dependency graph under the chosen threshold, and we empirically verify that it preserves predictive fidelity on representative tasks (see Table 2).

## F. Experiment Details

**Datasets** The investigated tasks probe specific capabilities of neural networks, such as arithmetic reasoning, variable tracking, and non-linear composition. We categorize them into three logical tiers based on their underlying complexity:

- **Linear Arithmetic.** This category involves linear transformations. *Non-state-tracking* tasks (e.g., `sum_last2`) require only local attention within a fixed window. In contrast, *State-tracking* tasks (e.g., `sum_last`) require the model to maintain a persistent memory state—implicitly simulating a finite state automaton (Zhang et al., 2025)—to compute the recurrent state over the sequence. Moreover, to evaluate generalization beyond pure arithmetic, we include tasks derived from classical mechanics (`freebody`, `gravity`, and `spring`), which are mathematically reducible to linear state-tracking tasks.

- **Non-Linear Composition.** These tasks necessitate non-linear activation logic. Variants range from local operations like `parity_last2` and `maximum_prev2` to global state-tracking tasks like `add_mod_3`. Success here requires the Numerical MLP to approximate non-linear functions (e.g., XOR) rather than simple linear aggregation.

- **Continuous Dynamics.** While the preceding two categories primarily focus on discrete variables, this category encompasses mathematical and physical tasks that require modeling and processing continuous-valued variables, such as `exponential_moving_average` and `free_fall_height`.

**Piecewise Linear Encoding.** For Numerical Attention, scalar variables selected by the constrained read matrices are first lifted to a higher-dimensional representation using Piecewise Linear Encoding (PLE). Specifically, we partition the numerical range of the training data into bins and learn an embedding vector for each bin boundary. A scalar value is then represented by linearly interpolating between the embeddings of its two neighboring boundaries.

**T5-style Relative Positional Biases.** In addition to content-based scores, each Numerical Attention head uses a T5-style relative positional bias. For every query-key pair, we compute the clipped relative offset between their sequence positions and map this offset to a learnable head-specific bias, which is added to the attention score before causal masking and sparse sampling.

**Training Details** All models adopt a decoder-only architecture implemented in PyTorch (Paszke et al., 2019), optimized via AdamW (Loshchilov & Hutter, 2019) to minimize MSE. The training dataset consists of $1,000,000$ samples. Unless otherwise specified, both the input and output sequences have a fixed length of 10. We employ a cosine annealing schedule to decay the learning rate from $0.05$ to $1 \times 10^{-6}$ over 50 epochs with a batch size of 512. (For the `freebody` and `gravity` tasks, models are trained for 100 epochs with an initial learning rate of 0.01.) To handle discrete optimization, we apply a geometric annealing schedule to the sampling temperature $\tau$, decreasing it from 10.0 to 0.1 to facilitate a gradual transition from continuous exploration to discrete selection. For each task, we conduct a grid search over the number of layers $\{1, 2\}$, attention heads $\{2, 3\}$, and sub-MLPs per layer $\{2, 3\}$. We run all experiments across three random seeds, reporting the mean performance and selecting the checkpoint with the best validation performance for subsequent analysis.

**Task Formulation** To facilitate algorithm extraction across varying positions, we design our task formulations to encourage the learning of position-invariant functions. Specifically, we categorize tasks according to their reliance on state-tracking: (1) Non-state-tracking tasks are formulated as Token-Tagging problems, requiring independent, element-wise predictions for each position. (2) State-tracking tasks are framed as Language Modeling problems, where the model performs autoregressive next-token prediction based on the input and historical context. To facilitate symbolic regression and align with the regression-oriented nature of these tasks, we adopt MSE as the training loss.

**Symbolic Regression Details** We apply PySR (Cranmer, 2023), a genetic-algorithm-based symbolic regression method, to identify the symbolic expression $\hat{f}$ that minimizes the prediction error on $\mathcal{D}$. Specifically, we use PySR with its default hyperparameter settings; moreover, for nonlinear tasks, we introduce additional operators such as ReLU. And to ensure reproducibility, we set the random seed to 42. The overall runtime for extracting an algorithm from a model ranges from a few minutes to several tens of minutes.

## G. Robustness of the Discrete Transformer

To evaluate the robustness of our framework under architectural variations, we investigate the impact of hyperparameters—specifically the number of layers, attention heads per layer, and sub-MLPs per layer—on both the convergence quality, measured by the MSE loss, and the complexity, quantified by the line count of the extracted program. We conduct experiments on the `sum_last2`, `parity_last2`, and `spring` tasks, varying the number of layers in $\{1, 2, 3, 4\}$, attention heads per layer in $\{0, 1, 2, 4, 8\}$, and sub-MLPs per layer in $\{0, 1, 2, 4, 8\}$.

Our results exhibit a clear threshold effect in expressive capacity. Once the architecture satisfies the minimal functional requirements of the target task (e.g., `sum_last2` requires at least one attention head to retrieve the token $x_{t-1}$), the Discrete Transformer reliably converges to a near-zero MSE loss (see Tables 3 and 4).

Furthermore, we find that over-parameterization has two distinct effects in our framework.

- First, excessive capacity increases the complexity of the extracted programs through structural redundancy: multiple modules may learn functionally equivalent behaviors, such as several attention heads retrieving the same offset token $x_{t-1}$. To examine whether this redundancy can be controlled, we introduce a simplicity bias by adding an $\ell_1$ regularization term with coefficient 0.001 on the linear output head in the over-parameterized setting with 4 layers, 4 heads per layer, and 4 sub-MLPs per layer. On the `sum_last2`, `parity_last2`, and `spring` tasks, this reduces the extracted line counts from 78 to 22, 85 to 42, and 96 to 34, respectively. At the same time, the extraction RMSE remains low under regularization ($1.01 \times 10^{-4}$, $8.71 \times 10^{-3}$, and $1.14 \times 10^{-4}$, respectively), suggesting that program complexity can be substantially reduced while maintaining high extraction fidelity.

- Second, the presence of redundant modules expands the search space and introduces noise during the exploration phase. Although this can complicate optimization and reduce training stability, we find that a slower temperature annealing schedule mitigates the issue by giving redundant components more time to resolve competition and converge to a valid discrete solution. For example, on the `spring` task, using the same over-parameterized model with 4 layers, 4 heads per layer, and 4 sub-MLPs per layer, extending the annealing horizon from 25 to 50 and 100 epochs progressively reduces the training MSE loss from $3.69 \times 10^{-3}$ to $2.35 \times 10^{-6}$ and further to $3.94 \times 10^{-7}$.

## H. Additional Synthesized Programs

In this section, we provide additional synthesized programs.

*Table 3.* Robustness tests across three tasks for architectures with one or two layers. For each task, SR denotes the success rate across three random seeds, Lines denotes the total lines of the extracted program, and RMSE denotes the mean RMSE across the three seeds. A dash indicates no valid program.

| Layers | Heads | sub-MLPs | sum_last2 | | | spring | | | parity_last2 | | |
|---|---|---|---|---|---|---|---|---|---|---|---|
| | | | SR | Lines | RMSE | SR | Lines | RMSE | SR | Lines | RMSE |
| 1 | 0 | 1 | 0/3 | – | $3.16 \times 10^{-1}$ | 0/3 | – | $1.25 \times 10^1$ | 0/3 | – | $5.00 \times 10^{-1}$ |
| 1 | 0 | 2 | 0/3 | – | $3.15 \times 10^{-1}$ | 0/3 | – | $1.25 \times 10^1$ | 0/3 | – | $5.00 \times 10^{-1}$ |
| 1 | 1 | 0 | 3/3 | 22 | 0 | 0/3 | – | $9.20 \times 10^0$ | 0/3 | – | $4.85 \times 10^{-1}$ |
| 1 | 1 | 1 | 3/3 | 25 | $4.02 \times 10^{-6}$ | 0/3 | – | $9.18 \times 10^0$ | 3/3 | 25 | $1.79 \times 10^{-7}$ |
| 1 | 1 | 2 | 3/3 | 22 | $9.67 \times 10^{-7}$ | 0/3 | – | $6.06 \times 10^0$ | 3/3 | 28 | $2.31 \times 10^{-3}$ |
| 1 | 2 | 0 | 3/3 | 22 | $5.25 \times 10^{-5}$ | 3/3 | 28 | 0 | 0/3 | – | $4.85 \times 10^{-1}$ |
| 1 | 2 | 1 | 3/3 | 22 | $3.12 \times 10^{-3}$ | 3/3 | 28 | $2.11 \times 10^{-9}$ | 3/3 | 31 | $2.26 \times 10^{-3}$ |
| 1 | 2 | 2 | 3/3 | 28 | $1.43 \times 10^{-6}$ | 3/3 | 28 | $7.41 \times 10^{-7}$ | 3/3 | 34 | $2.31 \times 10^{-3}$ |
| 1 | 2 | 4 | 3/3 | 34 | $8.85 \times 10^{-5}$ | 3/3 | 28 | $1.75 \times 10^{-8}$ | 3/3 | 39 | $3.77 \times 10^{-3}$ |
| 1 | 2 | 8 | 3/3 | 22 | $4.08 \times 10^{-6}$ | 3/3 | 28 | $3.69 \times 10^{-6}$ | 3/3 | 52 | $3.79 \times 10^{-4}$ |
| 1 | 4 | 2 | 3/3 | 22 | $1.84 \times 10^{-9}$ | 3/3 | 28 | $4.23 \times 10^{-6}$ | 3/3 | 39 | $1.53 \times 10^{-3}$ |
| 1 | 4 | 4 | 3/3 | 36 | $5.00 \times 10^{-7}$ | 3/3 | 28 | $9.90 \times 10^{-6}$ | 3/3 | 28 | $2.06 \times 10^{-3}$ |
| 1 | 4 | 8 | 3/3 | 31 | $3.34 \times 10^{-6}$ | 3/3 | 42 | $3.71 \times 10^{-5}$ | 3/3 | 54 | $3.36 \times 10^{-3}$ |
| 1 | 8 | 2 | 3/3 | 31 | $6.31 \times 10^{-7}$ | 3/3 | 34 | $1.30 \times 10^{-6}$ | 3/3 | 62 | $1.79 \times 10^{-3}$ |
| 1 | 8 | 4 | 3/3 | 52 | $8.21 \times 10^{-7}$ | 3/3 | 55 | $5.90 \times 10^{-3}$ | 3/3 | 55 | $2.11 \times 10^{-3}$ |
| 1 | 8 | 8 | 3/3 | 41 | $6.34 \times 10^{-6}$ | 3/3 | 71 | $6.85 \times 10^{-4}$ | 3/3 | 78 | $4.57 \times 10^{-3}$ |
| 2 | 2 | 2 | 3/3 | 47 | $8.27 \times 10^{-3}$ | 3/3 | 46 | $1.28 \times 10^{-3}$ | 3/3 | 47 | $3.48 \times 10^{-6}$ |
| 2 | 2 | 4 | 3/3 | 56 | $2.58 \times 10^{-3}$ | 3/3 | 53 | $5.52 \times 10^{-4}$ | 3/3 | 59 | $1.84 \times 10^{-3}$ |
| 2 | 2 | 8 | 3/3 | 53 | $1.74 \times 10^{-5}$ | 3/3 | 67 | $3.78 \times 10^{-4}$ | 3/3 | 52 | $1.10 \times 10^{-3}$ |
| 2 | 4 | 2 | 3/3 | 58 | $3.20 \times 10^{-4}$ | 3/3 | 65 | $2.34 \times 10^{-3}$ | 3/3 | 63 | $3.53 \times 10^{-3}$ |
| 2 | 4 | 4 | 3/3 | 60 | $1.31 \times 10^{-4}$ | 3/3 | 74 | $6.88 \times 10^{-3}$ | 3/3 | 59 | $1.39 \times 10^{-3}$ |
| 2 | 4 | 8 | 3/3 | 67 | $1.38 \times 10^{-4}$ | 3/3 | 75 | $2.03 \times 10^{-2}$ | 3/3 | 62 | $1.72 \times 10^{-3}$ |
| 2 | 8 | 2 | 3/3 | 73 | $9.81 \times 10^{-5}$ | 3/3 | 63 | $2.14 \times 10^{-3}$ | 3/3 | 80 | $9.58 \times 10^{-4}$ |
| 2 | 8 | 4 | 3/3 | 85 | $1.38 \times 10^{-4}$ | 3/3 | 75 | $8.02 \times 10^{-4}$ | 3/3 | 85 | $4.57 \times 10^{-3}$ |
| 2 | 8 | 8 | 3/3 | 72 | $1.09 \times 10^{-4}$ | 3/3 | 86 | $3.89 \times 10^{-3}$ | 3/3 | 68 | $3.44 \times 10^{-3}$ |

*Table 4.* Robustness tests across three tasks for architectures with three or four layers. The notation follows Table 3.

| Layers | Heads | sub-MLPs | sum_last2 | | | spring | | | parity_last2 | | |
|---|---|---|---|---|---|---|---|---|---|---|---|
| | | | SR | Lines | RMSE | SR | Lines | RMSE | SR | Lines | RMSE |
| 3 | 2 | 2 | 3/3 | 60 | $4.46 \times 10^{-5}$ | 3/3 | 53 | $4.05 \times 10^{-4}$ | 3/3 | 52 | $1.14 \times 10^{-4}$ |
| 3 | 2 | 4 | 3/3 | 55 | $1.54 \times 10^{-5}$ | 3/3 | 65 | $6.53 \times 10^{-3}$ | 3/3 | 66 | $4.41 \times 10^{-3}$ |
| 3 | 2 | 8 | 3/3 | 82 | $9.45 \times 10^{-4}$ | 3/3 | 82 | $4.86 \times 10^{-3}$ | 3/3 | 74 | $5.55 \times 10^{-4}$ |
| 3 | 4 | 2 | 3/3 | 71 | $3.71 \times 10^{-4}$ | 3/3 | 75 | $4.64 \times 10^{-3}$ | 3/3 | 58 | $1.48 \times 10^{-4}$ |
| 3 | 4 | 4 | 3/3 | 61 | $5.70 \times 10^{-5}$ | 3/3 | 82 | $2.12 \times 10^{-3}$ | 3/3 | 81 | $8.05 \times 10^{-5}$ |
| 3 | 4 | 8 | 3/3 | 82 | $1.92 \times 10^{-4}$ | 3/3 | 76 | $1.52 \times 10^{-3}$ | 3/3 | 79 | $4.62 \times 10^{-3}$ |
| 3 | 8 | 2 | 3/3 | 86 | $1.96 \times 10^{-3}$ | 3/3 | 92 | $7.65 \times 10^{-4}$ | 3/3 | 77 | $3.96 \times 10^{-3}$ |
| 3 | 8 | 4 | 3/3 | 85 | $4.90 \times 10^{-4}$ | 3/3 | 99 | $4.39 \times 10^{-3}$ | 3/3 | 70 | $3.48 \times 10^{-3}$ |
| 3 | 8 | 8 | 3/3 | 74 | $4.44 \times 10^{-4}$ | 3/3 | 101 | $1.05 \times 10^{-3}$ | 3/3 | 76 | $2.31 \times 10^{-3}$ |
| 4 | 2 | 2 | 3/3 | 64 | $1.84 \times 10^{-1}$ | 3/3 | 64 | $1.20 \times 10^{-3}$ | 3/3 | 68 | $2.45 \times 10^{-3}$ |
| 4 | 2 | 4 | 3/3 | 83 | $3.06 \times 10^{-4}$ | 3/3 | 86 | $1.27 \times 10^{-3}$ | 3/3 | 82 | $1.47 \times 10^{-4}$ |
| 4 | 2 | 8 | 3/3 | 108 | $1.40 \times 10^{-3}$ | 3/3 | 88 | $4.22 \times 10^{-3}$ | 3/3 | 62 | $4.50 \times 10^{-4}$ |
| 4 | 4 | 2 | 3/3 | 77 | $1.32 \times 10^{-4}$ | 3/3 | 64 | $5.28 \times 10^{-4}$ | 3/3 | 70 | $3.14 \times 10^{-3}$ |
| 4 | 4 | 4 | 3/3 | 86 | $1.11 \times 10^{-3}$ | 3/3 | 96 | $3.04 \times 10^{-3}$ | 3/3 | 85 | $3.26 \times 10^{-3}$ |
| 4 | 4 | 8 | 3/3 | 89 | $4.89 \times 10^{-4}$ | 3/3 | 124 | $9.06 \times 10^{-3}$ | 3/3 | 75 | $4.85 \times 10^{-3}$ |
| 4 | 8 | 2 | 3/3 | 78 | $1.33 \times 10^{-1}$ | 3/3 | 98 | $4.83 \times 10^{-4}$ | 3/3 | 79 | $4.45 \times 10^{-3}$ |
| 4 | 8 | 4 | 3/3 | 103 | $5.49 \times 10^{-4}$ | 3/3 | 116 | $2.40 \times 10^{-3}$ | 3/3 | 70 | $3.17 \times 10^{-3}$ |
| 4 | 8 | 8 | 3/3 | 123 | $5.83 \times 10^{-4}$ | 3/3 | 115 | $2.06 \times 10^{-3}$ | 3/3 | 102 | $2.32 \times 10^{-3}$ |

```python
import numpy as np
def maximum_prev2(input_seq):
    # V0_input
    input_arr = np.array(input_seq, dtype
    =float)
    seq_len = input_arr.shape[0]
    V0 = input_arr[:, 0]
    # V1_Attn_L0H0
    V1 = np.zeros(seq_len)
    # windowed max
    for t in range(seq_len):
        V1[t] = np.max(V0[max(0, t-1): t
    +1])
    # V2_Attn_L0H1
    V2 = np.zeros(seq_len)
    # windowed max
    for t in range(seq_len):
        V2[t] = np.max(V0[max(0, t-1): t
    +1])
    # Output Head
    output = 1.07 * V1 + -0.07 * V2
    return output
```

```python
import numpy as np
def minimum_prev2(input_seq):
    # V0_input
    input_arr = np.array(input_seq, dtype
    =float)
    seq_len = input_arr.shape[0]
    V0 = input_arr[:, 0]
    # V1_Attn_L0H0
    V1 = np.zeros(seq_len)
    # windowed min
    for t in range(seq_len):
        V1[t] = np.min(V0[max(0, t-1): t
    +1])
    # Output Head
    output = 1.00 * V1
    return output
```

```python
import numpy as np
def spring(input_seq):
    # V0_input
    input_arr = np.array(input_seq, dtype
    =float)
    seq_len = input_arr.shape[0]
    V0 = input_arr[:, 0]
    # V1_Attn_L0H0
    V1 = np.zeros(seq_len)
    # Fixed offset 9
    V1[9:] = V0[:-9]
    # V2_Attn_L0H1
    V2 = np.zeros(seq_len)
    # Fixed offset 10
    V2[10:] = V0[:-10]
    # V3_MLP_L0M0
    V3 = np.full(seq_len, 0.02)
    # V4_MLP_L0M1
    V4 = np.full(seq_len, -0.11)
    # V5_Attn_L1H0
    V5 = np.zeros(seq_len)
    # Fixed offset 2
    V5[2:] = V0[:-2]
    # V6_Attn_L1H1
    V6 = np.zeros(seq_len)
    # Fixed offset 2
    V6[2:] = V4[:-2]
    # V7_MLP_L1M0
    V7 = np.full(seq_len, -0.03)
    # Output Head
    output = 1.00 * V1 + 1.00 * V2 +
    -1.00 * V5 + 0.03 * V7 + -0.01 * V6
    return output
```

*Figure 6.* Intervened synthesized programs revealing alternative logical pathways. Left: When MLP-based arithmetic is prohibited, the model solves maximum_prev2 and minimum_prev2 by shifting to a pure attention mechanism. The extracted code shows explicit *Windowed Extrema* attention patterns. Right: For the spring task, intervention in the information flow (masking recent history) forces the model to learn a high-order recurrence relation, validating the model's ability to uncover multiple equivalent algorithms for the same data distribution.

```python
import numpy as np
def sum_last(input_seq):
    # V0_input
    input_arr = np.array(input_seq, dtype
    =float)
    seq_len = input_arr.shape[0]
    V0 = input_arr[:, 0]
    # V1_Attn_L0H0
    V1 = np.zeros(seq_len)
    # Fixed offset 9
    V1[9:] = V0[:-9]
    # Output Head
    output = 1.00 * V0 + 1.00 * V1
    return output
```

```python
import numpy as np
def bitwise_and(input_seq):
    # V0_input, V1_input
    input_arr = np.array(input_seq, dtype
    =float)
    seq_len = input_arr.shape[0]
    V0 = input_arr[:, 0]
    V1 = input_arr[:, 1]
    # V4_MLP_L0M0
    V4 = np.full(seq_len, 0.02)
    # V5_MLP_L0M1
    V5 = ((((V0 - 0.49) * 3.28) / (V1 +
    -0.52)) - 3.06) * 0.19
    # Output Head
    output = 0.49 * V1 + 0.48 * V0 + 0.40
     * V5 + 0.02 * V4
    return output
```

```python
import numpy as np
def add_mod_3(input_seq):
    # V0_input
    input_arr = np.array(input_seq, dtype
    =float)
    seq_len = input_arr.shape[0]
    V0 = input_arr[:, 0]
    # V1_Attn_L0H0
    V1 = np.zeros(seq_len)
    # Fixed offset 9
    V1[9:] = V0[:-9]
    # V2_Attn_L0H1
    V2 = np.zeros(seq_len)
    # Fixed offset 9
    V2[9:] = V0[:-9]
    # V3_MLP_L0M0
    V3 = ((-0.70 / (((V0 + V1) + 0.59) *
    -0.56)) - 1.69) + (1.05 / (V0 + (V1 -
     2.53)))
    # Output Head
    output = -0.53 * V3 + 0.47 * V2 +
    -0.32 * V1 + 0.16 * V0
    return output
```

*Figure 7.* Synthesized programs for sum_last (Top Left), bitwise_and (Bottom Left), and add_mod_3 (Right).

```python
import numpy as np
def freebody(input_seq):
    # V0_input
    input_arr = np.array(input_seq, dtype
    =float)
    seq_len = input_arr.shape[0]
    V0 = input_arr[:, 0]
    # V1_Attn_L0H0
    V1 = np.zeros(seq_len)
    # Fixed offset 9
    V1[9:] = V0[:-9]
    # V2_Attn_L0H1
    V2 = np.zeros(seq_len)
    # Fixed offset 10
    V2[10:] = V0[:-10]
    # V4_MLP_L0M1
    V4 = -0.26*V0 + 0.04
    # V5_MLP_L0M2
    V5 = -0.21*V0 + -0.04
    # V6_Attn_L1H0
    V6 = np.zeros(seq_len)
    # Fixed offset 2
    V6[2:] = V4[:-2]
    # V7_Attn_L1H1
    V7 = np.zeros(seq_len)
    # Fixed offset 2
    V7[2:] = V5[:-2]
    # Output Head
    output = 1.19 * V0 + 1.06 * V7 + 1.06
     * V6 + 1.00 * V1 + -0.65 * V4 +
    -0.65 * V5 + 0.50 * V2
    return output
```

```python
import numpy as np
def gravity(input_seq):
    # V0_input
    input_arr = np.array(input_seq, dtype
    =float)
    seq_len = input_arr.shape[0]
    V0 = input_arr[:, 0]
    # V1_Attn_L0H0
    V1 = np.zeros(seq_len)
    # Fixed offset 9
    V1[9:] = V0[:-9]
    # V2_Attn_L0H1
    V2 = np.zeros(seq_len)
    # Fixed offset 10
    V2[10:] = V0[:-10]
    # V4_MLP_L0M1
    V4 = 0.00
    # V5_MLP_L0M2
    V5 = -0.40*V0 + -3.05
    # V6_Attn_L1H0
    V6 = np.zeros(seq_len)
    # Fixed offset 2
    V6[2:] = V5[:-2]
    # V7_Attn_L1H1
    V7 = np.zeros(seq_len)
    # Fixed offset 2
    V7[2:] = V4[:-2]
    # Output Head
    output = 1.24 * V6 + 1.20 * V0 + 1.00
     * V1 + -0.74 * V5 + 0.50 * V2 +
    -0.01 * V4 + 0.01 * V7
    return output
```

```python
import numpy as np
def spring(input_seq):
    # V0_input
    input_arr = np.array(input_seq, dtype
    =float)
    seq_len = input_arr.shape[0]
    V0 = input_arr[:, 0]
    # V1_Attn_L0H0
    V1 = np.zeros(seq_len)
    # Fixed offset 9
    V1[9:] = V0[:-9]
    # V2_Attn_L0H1
    V2 = np.zeros(seq_len)
    # Fixed offset 1
    V2[1:] = V0[:-1]
    # V4_MLP_L0M1
    V4 = (V2 + ((V1 - ((V1 + 2.26) + (V2
    - 1.54))) * 2.00)) + (V2 + 1.43)
    # Output Head
    output = -1.00 * V2 + 1.00 * V0 +
    1.00 * V1 + 0.04 * V4
    return output
```

*Figure 8.* Synthesized programs for `freebody` (Top Left), `spring` (Bottom Left), and `gravity` (Right).

*Table 5.* Performance and parameters of the Discrete Transformer on the algorithmic benchmark. The columns Layers, Heads, and sub-MLPs correspond to the number of layers, the number of attention heads per layer, and the number of sub-MLPs per layer, respectively. Loss values are averaged MSE over three random seeds.

| Task Name | Description | Layers | Heads | sub-MLPs | Train Loss |
|---|---|---|---|---|---|
| *Linear Arithmetic* | | | | | |
| sum_last2 | Sum of the last two numbers | 1 | 2 | 2 | $8.82 \times 10^{-9}$ |
| diff_last2 | Difference of the last two numbers | 1 | 2 | 2 | $6.59 \times 10^{-16}$ |
| sum | Cumulative sum of the sequence | 1 | 2 | 2 | $1.71 \times 10^{-14}$ |
| div_3 | Binary division by 3 | 1 | 2 | 2 | $2.20 \times 10^{-1}$ |
| freebody | Simulate freebody dynamics | 2 | 2 | 3 | $9.57 \times 10^{-11}$ |
| gravity | Simulate gravity dynamics | 2 | 2 | 3 | $2.34 \times 10^{-11}$ |
| spring | Simulate spring dynamics | 1 | 2 | 2 | $1.41 \times 10^{-7}$ |
| magnetic | Simulate magnetic dynamics | 2 | 2 | 3 | $1.12 \times 10^{2}$ |
| *Non-Linear Composition* | | | | | |
| parity_last2 | Parity of the last two numbers | 1 | 2 | 2 | $4.87 \times 10^{-13}$ |
| maximum_prev2 | Maximum of the last two numbers | 1 | 2 | 2 | $1.11 \times 10^{-6}$ |
| minimum_prev2 | Minimum of the last two numbers | 1 | 2 | 2 | $1.64 \times 10^{-5}$ |
| unique_prev2 | 1 if last two numbers are equal | 1 | 2 | 2 | $4.19 \times 10^{-14}$ |
| bitwise_and | Bitwise AND | 1 | 2 | 2 | $1.10 \times 10^{-18}$ |
| bitwise_or | Bitwise OR | 1 | 2 | 2 | $5.66 \times 10^{-4}$ |
| bitwise_not | Bitwise NOT | 1 | 2 | 2 | $1.44 \times 10^{-17}$ |
| bitwise_xor | Bitwise XOR | 1 | 2 | 2 | $6.74 \times 10^{-20}$ |
| abs | Abs value of the number | 1 | 2 | 2 | $2.69 \times 10^{-14}$ |
| abs_of_diff | Abs difference of last two numbers | 1 | 2 | 2 | $9.72 \times 10^{-9}$ |
| parity | Cumulative parity | 1 | 2 | 2 | $8.15 \times 10^{-17}$ |
| parity_zeros | Cumulative parity of number of zeros | 1 | 2 | 2 | $2.25 \times 10^{-8}$ |
| add_mod_3 | Cumulative sum modulo 3 | 1 | 2 | 2 | $9.72 \times 10^{-13}$ |
| bit_dot_prod_mod_2 | Cumulative dot product of bits mod 2 | 1 | 2 | 2 | $5.55 \times 10^{-4}$ |
| bit_addition | Binary addition | 2 | 4 | 4 | $2.49 \times 10^{-2}$ |
| balanced_parenthesis | 1 if parentheses are balanced | 1 | 2 | 2 | $2.08 \times 10^{-2}$ |
| *Continuous Dynamics* | | | | | |
| exponential_moving_average | Exponential moving average | 1 | 2 | 2 | $1.62 \times 10^{-13}$ |
| free_fall_height | Free-fall height | 1 | 1 | 1 | $8.53 \times 10^{-5}$ |
| linear_drop | Linear drop | 2 | 2 | 2 | $2.52 \times 10^{-7}$ |
| quadratic_drop | Quadratic drop | 2 | 2 | 2 | $8.74 \times 10^{-7}$ |
| damped_harmonic_oscillator | Damped harmonic oscillator | 1 | 2 | 2 | $8.84 \times 10^{-8}$ |

