# OpenReview forum: "Weights to Code: Extracting Interpretable Algorithms from the Discrete Transformer"
_ICML.cc/2026/Conference — ICML 2026 regular_

### Official Review · Reviewer_Dq6Z · 2026-02-25

**Soundness:** 3
**Presentation:** 3
**Significance:** 3
**Originality:** 4
**Overall Recommendation:** 5
**Confidence:** 2

**Summary:**

The paper proposes a new neural network architecture (the Discrete Transformer) that is designed to facilitate the algorithms that it learns. The architecture modifies the Transformer such that layers directly operate on the raw values of variables instead of operating on embeddings, and introduces a mechanism to gradually learn more discrete operations during training (e.g., hard attention instead of soft attention). They also introduce a method that iteratively decodes the algorithms from the Transformer by classifying attention patterns and using symbolic regression.  Experiments indicate the approach is effective for learning and recovering simple arithmetic algorithms, and is better suited than SOTA for problems with continuous variables.

**Compliance With Llm Reviewing Policy:**

Affirmed.

**Final Justification:**

The paper stands out in its originality: the authors propose a new neural network architecture with inductive priors that facilitate reverse engineering the algorithms that are encoded in its weights. There is also novelty in the method they use to decode such algorithms.

Experiments indicate it that it compares favorably with prior art (i.e., MIPS) on 'Continuous Dynamic' algorithms, and performs similarly on other problem types.

My initial concerns regarding soundness and presentation have be addressed in the rebuttal:
- The authors committed to adding clarifications and highlighting limitations where appropriate.
- My concerns about the effect on over-parametrization on the complexity of extracted algorithm have been addressed with an additional regularization experiment and the fact that the authors will discuss it as a potential challenge in the main body of the paper.

A weaker point of the paper is its evaluation scope: the experiments were done on rather simple arithmetic problems. The immediate practical impact on algorithm extraction therefore remains unclear. Considering the novelty of the approach, I do not see this as a major weakness as long as the claims of the paper are not overstated.

**Key Questions For Authors:**

1. I would like to see more discussion/analysis on why the attention heads reduce to two seemingly simple patterns (line 194, right column). Would these patterns be sufficient for more complicated algorithms?

With respect to the over-parametrization issue discussed in Appendix D line 631:

2. Do you have empirical evidence for your claim that the issue can be mitigated by adopting a slower temperature annealing schedule?

3. Provided that the required capacity is problem-dependent, is there a way to determine an appropriate schedule in advance?

**Limitations:**

The authors could discuss the challenges with respect to over-parametrization in the main body of the paper. I believe this issue is very relevant when applying the approach to larger problem sizes.

I also encourage authors to acknowledge that the experiments are done on simple problems for which small model sizes suffice. I believe the number of layers, attention heads, MLP dimensions is only mentioned in appendix, so readers might initially miss this fact. Any insights related to the applicability of the architecture (components) to more complicated problems would be interesting as well.

**Edit**: I am content with the modifications that the author propose in their rebuttal.

**Strengths And Weaknesses:**

**Strengths**:
- [Soundness] The architecture is well-thought-out and experiments validate its potential.
- [Originality] To my the best of my knowledge, the architecture modifications which are proposed and the ideas behind them are substantially different from existing architectures.
- [Significance] I agree with the authors that algorithm extraction could have the potential to discover innovative algorithms.
- [Significance]  The paper presents an architecture that is easier to reverse engineer, which is useful from mechanistic interpretability.

**Weaknesses**:
- [Soundness/Significance] The method is only evaluated on toy problems.
- [Soundness] I would like to see more discussion/analysis on why the attention heads reduce to two seemingly simple patterns (line 194, right column). It's not clear whether this inherent to Discrete Transformers or an artifact of evaluating on simple problems.
- [Soundness/Presentation] The fact that over-parameterized architectures produce complicated programs (as discussed in Appendix D line 631) does seem like an important limitation and is not mentioned in the main body of the paper. The authors claim that the issue can be mitigated by adopting slower temperature annealing schedule. I would like to see more analysis on this.
- [Presentation] Section 3 could be more explicit on how information is processed in the time/sequence axis. I assume that h_{l+1} corresponds to representation of one sequence position/timestep and that equations 1-3 happen in parallel for all timesteps, but given that this is a new architecture this is not obvious. A small architecture diagram might help.

Minor comments:
- The paper would be more self-contained by explaining the architectural building blocks they use Piecewise Linear Encoding, T5-style relative positional biases, etc.
- Figures 5 and 6 are difficult to read.

---

> ### Author Rebuttal · Authors · 2026-03-30
>
> Thank you for the thoughtful and constructive review. We sincerely appreciate your comments and respond to each point below.
>
> On the "toy problems" concern.
> We agree that the current evaluation is benchmark-scale with small model sizes, and we will clarify this more explicitly in the main text. Our goal in this paper is not to claim immediate scalability to large-scale algorithm discovery, but to establish a Transformer architecture from which executable algorithms can be extracted with high fidelity. For this goal, we include "toy problems" with known underlying rules. Importantly, however, these are not confined to a single problem class; rather, the benchmark covers a wide range of settings, including linear and non-linear arithmetic, physical dynamics, and continuous dynamics.
>
> On applicability beyond the current benchmark.
> Our current results suggest that this architecture is most naturally suited to problems whose computation can be decomposed into sparse variable routing and low-arity state updates. Such problems include longer-horizon recurrences, simulation-style updates, and multi-stage symbolic transformations, where complexity arises from the composition of simple operations rather than from a single highly expressive black-box module. More broadly, the Discrete Transformer deliberately trades some expressivity for interpretability: router-style attention emphasizes interpretable selection/routing rather than dense feature mixing, while the MLP performs bounded-arity updates on explicit scalar variables. We do not claim that the current formulation is universally applicable to all complex tasks, particularly those that require highly entangled feature mixing or very high-arity interactions. We will clarify this scope of applicability and discuss its limitations more explicitly.
>
> On the two seemingly simple patterns.
> Our interpretation is that their prominence appears to be partly architectural and partly benchmark-dependent: the Discrete Transformer biases attention toward router-style selection, while arithmetic computation is delegated to the MLP; on our current tasks the active head behaviors naturally collapse to simple routing patterns. However, more complex algorithms need not be limited to these two patterns. For more complex tasks, if attention is not well-approximated by our current classes, the pipeline can be extended by adding new head classes and fitting unmatched but causally important heads. Some remain router-like (e.g., retrieval/equality-match), while others are genuinely non-router computational heads (e.g., mean/weighted aggregation; for these we retain soft attention rather than hardening it). As diagnostics, we additionally tested two such cases: on *retrieve_value_by_key*, adding a retrieval head yields extraction RMSE $1.28\times10^{-10}$; on *prefix_mean*, adding a mean head yields RMSE $3.14\times10^{-4}$.
>
> On over-parameterization and the annealing schedule.
> We believe there is a slight misunderstanding here. In Appendix D, "this issue" refers specifically to the discrete-search instability caused by redundant modules, rather than the increase in program complexity. Our claim is that a slower temperature annealing schedule helps redundant components resolve competition and converge to a valid discrete solution; and we have added a direct ablation on *spring* with an over-parameterized model (layers: 4, heads per layer: 4, sub-MLPs per layer: 4): using longer annealing horizons (25, 50, and 100 epochs) reduce the MSE loss from $3.69\times10^{-3}$ to $2.35\times10^{-6}$ and then to $3.94\times10^{-7}$. We will clarify the wording and move this point to the main text.
>
> Regarding schedule selection, we do not claim that an optimal schedule can be determined exactly in advance. Our guidance is therefore heuristic: if the soft objective has largely converged but the hard objective remains poor during annealing, a more conservative (slower) annealing schedule is worth trying.
>
> On the sequence/time axis in Section 3. Yes, each sequence position $t$ maintains its own numerical residual stream $h_l(t)$. Equation (1) concatenates newly produced scalar variables to the residual stream of that same position; Equation (2) selects scalar operands from that position’s stream; and Equations (3)–(5) are evaluated in parallel over all query positions, with attention being the only component that reads across the sequence axis. By contrast, the Numerical MLP acts position-wise. We agree that this should be stated much more explicitly, and we will add a small architecture diagram to make the information flow over the variable axis vs. the sequence axis visually clear.
>
> On minor presentation comments. We agree that the paper can be made more self-contained by adding more details explaining PLE and T5-style relative positional biases. We also agree that Figures 5 and 6 should be redesigned to improve readability. We appreciate these suggestions and will incorporate them into the revision.

---

> > ### Author Rebuttal · Reviewer_Dq6Z · 2026-04-01
> >
> > I appreciate the authors' detailed response. I believe the suggested modifications and additional experiments will improve the paper.
> >
> > I still have one follow-up question:
> > > On over-parameterization and the annealing schedule. We believe there is a slight misunderstanding here. In Appendix D, "this issue" refers specifically to the discrete-search instability caused by redundant modules, rather than the increase in program complexity. Our claim is that a slower temperature annealing schedule helps redundant components resolve competition and converge to a valid discrete solution
> >
> > I am a bit confused about this comment. Appendix D, line 633 reads: "Specifically, excessive capacity **tends to inflate the length of the extracted programs due to structural redundancy**: multiple modules may learn functionally equivalent behaviors (e.g., several attention heads learning the same retrieval pattern for $x_{t−1}$)". Does this not mean that the complexity of the extracted problem increases due to over-parametrization? Can the authors clarify?
> >
> > **Edit**: The authors addressed my last question in their comment below. In light of the changes that the authors suggested and the additional results they reported, I will update my scores as follows:
> > - Soundness from 2 to 3
> > - Overall Recommendation from '4 Weak accept' to '5 Accept'.

---

> > > ### Author Response · Authors · 2026-04-01
> > >
> > > Thank you for the follow-up — you are correct, and our previous wording was imprecise.
> > >
> > > Appendix D does indeed state that over-parameterization can increase the complexity of the extracted program. More precisely, we intended to distinguish between two related effects of structural redundancy in the over-parameterized regime:
> > > - an interpretability/complexity effect, where multiple modules learn functionally equivalent behaviors and thereby inflate the length of the extracted program; and
> > > - an optimization/discretization effect, where the same redundancy makes the soft-to-hard transition harder because redundant modules compete during annealing.
> > >
> > > Our point about slower temperature annealing referred **only to the second effect**: it can help redundant components resolve competition and converge to a valid discrete solution more reliably, but it does not by itself guarantee a compact extracted program. So we agree with your reading that Appendix D also identifies increased extracted-program complexity as a genuine consequence of over-parameterization.
> > >
> > > We also agree that this is an important limitation, especially when considering scaling to larger problem sizes, and we will move this discussion into the main text and strengthen it with additional empirical analysis. Beyond the annealing ablation, we have tested a more direct simplicity bias by adding an L1 regularization term (coefficient 0.001) on the linear output head in the over-parameterized setting (4 layers, 4 heads per layer, and 4 sub-MLPs per layer). On *sum_last2*, *parity_last2*, and *spring*, this reduced the extracted line counts substantially, from 78 to 22, 85 to 42, and 96 to 34, respectively. Meanwhile, the extraction RMSE remained low in absolute terms under regularization ($1.01 \times 10^{-4}$, $8.71 \times 10^{-3}$, and $1.14 \times 10^{-4}$, respectively), suggesting that a substantial reduction in extracted-program complexity can be achieved without a commensurate loss in extraction fidelity.
> > >
> > > In the revision, we will make this distinction explicit: slower annealing mainly addresses discretization stability, whereas keeping the extracted program compact may require controlling model capacity and/or adding explicit simplicity bias. We will also move this discussion from Appendix D into the main text so that this limitation is visible to readers rather than easy to miss.
> > >
> > > **We truly appreciate you taking the time to provide this additional feedback, and we hope our clarification can address the concerns you raised. Thanks for your invaluable feedback, as we strive to improve the quality and impact of our work!**

---

### Official Review · Reviewer_dDTH · 2026-03-09

**Soundness:** 2
**Presentation:** 3
**Significance:** 3
**Originality:** 3
**Overall Recommendation:** 5
**Confidence:** 4

**Summary:**

This paper introduces the *Discrete Transformer*, a Transformer-like architecture designed to reduce representation entanglement and make learned algorithms more amenable to symbolic recovery. The core contribution is an *automated extraction pipeline* that (i) identifies a small set of recurring attention-head behaviors that act as deterministic routing primitives, (ii) fits closed-form expressions for sub-MLPs via symbolic regression, and (iii) reconstructs an *executable end-to-end program* by backward traversal from the output head, with optional symbolic simplification. Experiments on a suite of algorithmic tasks show that extracted programs can match model behavior with high fidelity, and that architectural inductive biases provide control over the form of the synthesized program.

**Compliance With Llm Reviewing Policy:**

Affirmed.

**Final Justification:**

This paper presents a well-motivated and original architecture-plus-extraction framework for recovering executable programs from learned neural computation. I found the work technically solid on its core empirical claim: within the proposed constrained architecture, the extraction pipeline can recover runnable symbolic programs with high fidelity, and the examples are clear and compelling. The main strengths are the coherence of the end-to-end methodology, the tangible interpretability outcome, and the broader significance of pursuing executable algorithm extraction.

My original concerns were mainly about claim framing, scaling/expressivity limitations, and support for some key design choices. The rebuttal addressed these concerns well. In particular, the authors agreed to revise the overstated “rigorous theoretical analysis” wording, clarified the scope of the correctness claim, explicitly acknowledged the expressivity/scaling tradeoff induced by bounded-fan-in operators and router-style attention, and pointed to/additionally provided capacity and arity ablations that improve confidence in the empirical story. They also clarified that the reported attention head types are not intended as a complete taxonomy and that the recovery pipeline can be extended with additional head classes when needed.

Overall, I view the paper as sound, reasonably original, and significant for the mechanistic interpretability / algorithm extraction community, with clear presentation and a useful contribution despite the remaining scope limitations. The rebuttal increased my confidence in the work and changed my evaluation from **4 (Weak Accept)** to **5 (Accept)**.

**Key Questions For Authors:**

1. “Rigorous theoretical analysis”: What theoretical properties are claimed precisely? Can the authors provide at least one formal statement (even under simplifying assumptions) about correctness/identifiability of component recovery and backward reconstruction, or revise the wording to reflect a primarily empirical analysis?

    *Impact*: A clear theory claim or toned-down framing would improve soundness.

2. Capacity and robustness: Can the authors report systematic ablations across layers/heads/width (including under- and over-capacity), with extraction success rate and fidelity (RMSE/accuracy) across multiple seeds?

    *Impact*: Would directly support (or refine) the “minimal requirements ⇒ consistent extraction” claim and increase confidence.

3. Attention beyond router regimes.

    The extraction pipeline assumes attention heads behave as deterministic routers (fixed-offset / windowed-extrema), and the paper reports that these two types dominate in the studied settings. However, in other algorithmic regimes attention can take on non-router computational roles (e.g., content-based retrieval or multiplicative/operator-like mechanisms; see Nanda et al., 2023; Zhong et al., 2023).

    *Question*: Please add a discussion (and ideally a small diagnostic experiment) on how the recovery pipeline would handle cases where attention is not well-approximated as a router—what additional head classes would be needed, how they would be identified, and how incorporating them would affect the backward reconstruction into an executable program.

4. Circuit-complexity / scaling clarification (see Weakness #4).

    *Question:* Please add an explicit discussion/limitations statement connecting the Discrete Transformer’s discretization + bounded-fan-in operator design (and router-style attention) to **input-length–dependent depth requirements** (e.g., $\Theta(\log n)$ for reductions, potentially $\Theta(n)$ for scan/state propagation), and clarify which task families are expected to degrade unless depth scales with sequence length (e.g., *div3* as a remainder-tracking scan).

5. Arity (k) choice: Why fix sub-MLPs to (k=2)?

    *Impact*: Clarifies sensitivity and scalability.

**Limitations:**

No, please see Weakness#4.

**Strengths And Weaknesses:**

## Strengths
1. Clear problem and strong motivation: executable algorithm extraction is an important goal for mechanistic interpretability and demonstration-free discovery.

2. Cohesive pipeline: the paper presents a practical end-to-end methodology (component identification → backward reconstruction → simplification) and demonstrates it on multiple tasks.

3. Interpretability outcome is tangible: extracted artifacts are runnable and often compact, which is a meaningful step beyond qualitative “attention pattern” analysis.

4. Architecture/extraction co-design: the work makes a coherent case that architectural constraints can improve extractability; the observed exploration→exploitation annealing dynamic is interesting and potentially useful.

## Weaknesses

1. **Over-claiming “rigorous theoretical analysis.”**

The paper states it provides a rigorous analysis of theoretical and empirical properties, but the current version reads primarily as an empirical/hypothesis-testing + symbolic-regression pipeline with several heuristic thresholds/configuration choices. I did not see a corresponding formal analysis (e.g., identifiability/correctness guarantees, robustness bounds, or scaling/complexity characterization) supporting the strength of this claim.

2. **Limited attention taxonomy and unclear generalization beyond router heads.**

The extraction pipeline hinges on attention heads behaving as deterministic routers (primarily fixed-offset and windowed-extrema). While the paper notes other head behaviors may exist but are “mostly noise,” the explored models are small and the task suite appears relatively simple, making it unclear whether this taxonomy reflects a general property of the Discrete Transformer or is largely an artifact of the particular benchmark family.

In particular, it would help to clarify why the dominant recovered head types are exactly these two: are they

(i) simply the patterns that emerge for the specific tasks studied, in which case a discussion/empirical probe of how the taxonomy changes for more general problem classes is needed, or

(ii) deliberately chosen to yield a “router + operator” instruction set with some form of algorithmic completeness (and if so, why is windowed-extrema necessary given fixed-offset routing plus additional operator depth could in principle implement local reductions)?

More broadly, in many algorithmic regimes attention can act as a computational operator (e.g., content-based retrieval or multiplicative interactions) rather than a pure pointer router, as observed in mechanistic studies of modular arithmetic settings (Nanda et al., 2023; Zhong et al., 2023). Note this comment is not that a router+operator system cannot in principle reproduce such tasks; rather, the concern is that **training may naturally allocate attention to non-router roles, and it is unclear how the current recovery procedure would extend or remain robust when attention deviates from the router assumption.**

Nanda et al. (2023): Neel Nanda, Lawrence Chan, Tom Lieberum, Jess Smith, and Jacob Steinhardt. Progress measures for grokking via mechanistic interpretability. In The Eleventh International Conference on Learning Representations (ICLR), 2023.

Zhong et al. (2023): Ziqian Zhong, Ziming Liu, Max Tegmark, and Jacob Andreas. The clock and the pizza: Two stories in mechanistic explanation of neural networks. Advances in Neural Information Processing Systems (NeurIPS), 36:27223–27250, 2023.

3. **“Minimal functional requirements ⇒ consistent convergence and extraction” is not well supported.**

The paper suggests that once the architecture meets minimal functional requirements, training reliably reaches near-zero MSE and yields executable extracted programs. This reads like a phase-transition claim, but the evidence is limited: I did not see systematic ablations showing both under-capacity failure cases and over-capacity regimes (the latter being common in practice).

4. **Expressivity vs. circuit complexity deserves a clearer, explicit limitation statement.**

While the paper acknowledges a trade-off in expressivity, I think this needs to be made explicit because it is not a constant-factor cost: the proposed discretization + low-arity operators can induce an input-length–dependent depth gap. Conventional transformers (with global attention aggregation) are often informally associated with **constant-depth** global mixing / TC(^0)-like behavior for some global functions, whereas if attention is constrained primarily to routing and sub-MLPs have **bounded fan-in** (e.g., arity-2), then many global **computations require depth that grows with input length**—typically $\Theta(\log n)$ for parallel reductions and potentially $\Theta(n)$ for scan/state-propagation patterns. Given the *“Discrete Transformer”* framing, I recommend the paper explicitly warn readers about this shift in asymptotic depth requirements as sequence length increases, and clarify which task families are expected to degrade unless depth scales accordingly.

The current results are consistent with this concern: for example, if “div3” corresponds to binary division by 3, it is naturally a scan/state-tracking transduction (tracking a small remainder across positions), which is exactly the kind of computation that tends to be depth/step-hungry under bounded-fan-in router-style computation.

5. **Key design choices (e.g., (k=2) sub-MLP arity) are not ablated.**

The fixed choice (k=2) seems central to both extraction tractability and architectural bias, but it is not empirically justified via ablations over (k). This leaves uncertainty about how sensitive the approach is and how it should scale to larger models and more complex tasks.

---

> ### Author Rebuttal · Authors · 2026-03-29
>
> Thanks for your detailed and constructive review. We sincerely appreciate these comments and respond point by point in the order of the weaknesses.
>
> W1:
> We agree that the current wording overstates the theoretical contribution, and will revise “rigorous theoretical analysis” to “empirical analysis”. Our claim is not universal identifiability of arbitrary circuits, but a restricted correctness result: for a trained Discrete Transformer on domain $\mathcal{X}$, if (i) selectors and active heads converge to deterministic one-hot choices, (ii) each active head is uniquely identified within our tested hypothesis class on $\mathcal{X}$, (iii) each active MLP is exactly recoverable in the symbolic class on $\mathcal{X}$, and (iv) pruning preserves all active variables, then backward traversal returns a symbolic program $P$ with $P(x)=f_{\mathrm{DT}}(x)$ for all $x\in\mathcal{X}$, up to algebraic equivalence and redundant inactive components; we do not claim uniqueness in over-parameterized settings.
>
> W2:
> We do not claim that fixed-offset and windowed-extrema form a complete taxonomy of attention behaviors; they are benchmark- and inductive-bias-dependent. Nor do we claim windowed-extrema is necessary for completeness: local reductions could also be implemented by fixed-offset routing plus deeper operators, but windowed-extrema gives a more compact description of the learned solution (Section 6.2).
>
> More generally, if attention is not well-approximated by our current router classes, the pipeline can be extended by adding new head classes and fitting unmatched but causally important heads. Some remain router-like (e.g., retrieval/equality-match), while others are genuinely non-router computational heads (e.g., mean/weighted aggregation). Backward reconstruction then remains largely unchanged: router-like heads are mapped to routers, while non-router heads are treated as computational operators. As diagnostics, we additionally tested two such cases: on retrieve_value_by_key, adding a retrieval head yields extraction RMSE $1.28\times10^{-10}$; on prefix_mean, replacing hard attention with soft attention and adding a mean head yields RMSE $3.14\times10^{-4}$.
>
> W3:
> We agree that this point is not sufficiently explicit in the current presentation. In fact, it is already partially addressed in Appendix D, where we conduct systematic capacity sweeps over Layers, Heads, and sub-MLPs on sum_last2, parity_last2, and spring, covering both under- and over-capacity settings. Beyond Fig. 6, we also include corresponding results for spring and parity_last2 (https://anonymous.4open.science/r/anonymous-data-9D46/), reporting extraction success rate and fidelity over three random seeds. We appreciate that this evidence is currently easy to miss, and will revise the manuscript to surface it more clearly.
>
> W4:
> Thanks for highlighting this important limitation—we fully agree. Our Discrete Transformer intentionally trades expressivity for interpretability: router-style attention primarily performs selection/routing rather than dense feature mixing, and each sub-MLP has bounded fan-in. An $L$-layer model is therefore better viewed as a selector-augmented bounded-depth computation graph, where arithmetic aggregation across many positions can expand only gradually with depth. This induces an input-length-dependent depth requirement: global reductions generally require depth that grows with input length (e.g., $O(\log n)$ under hierarchical composition), while scan/state-propagation patterns may require $O(n)$ depth unless the task admits an associative hierarchical summary. We will make this limitation explicit and clarify that, at fixed depth, the architecture is best suited to selector-style retrieval, fixed-window composition, and fixed-order recurrences, whereas tasks involving global aggregation, carry/remainder propagation, or automaton-/stack-like transductions become more challenging as sequence length grows.
>
> W5:
> We agree that sub-MLP arity ($k$) is a key design parameter. Our choice ($k=2$) is intentional: it is the smallest arity that captures many benchmark primitives while keeping symbolic regression tractable and component recovery stable. New ablations clarify this trade-off. For a small model (Layers=2, Heads=2, sub-MLPs=2), $k=1$ underfits add_mod_3 / bit_dot_prod_mod_2 (RMSE $6.46\times10^{-2}$ / $2.36\times10^{-1}$); $k=2$ already reaches $3.10\times10^{-4}$ / $3.00\times10^{-4}$. Larger $k$ does not uniformly help ($k=4$: $2.98\times10^{-3}$ / $8.44\times10^{-4}$). For parity_last3, which requires genuine 3-way interaction, $k=3$ is best ($3.33\times10^{-6}$), while $k=2$ fails ($4.69\times10^{-1}$) and $k=4$ also succeeds ($2.06\times10^{-3}$). For a larger model (L=4, H=4, M=4), increasing $k$ from 2 to 4 worsens add_mod_3 ($3.69\times10^{-3} \to 7.75\times10^{-3}$) and bit_dot_prod_mod_2 ($2.59\times10^{-3} \to 1.93\times10^{-2}$), while only modestly improving parity_last3 ($1.31\times10^{-2} \to 6.17\times10^{-3}$).

---

> > ### Author Rebuttal · Reviewer_dDTH · 2026-04-02
> >
> > The rebuttal adequately addresses my main concerns and increases my confidence in the paper. In particular, the authors agree to revise the overstated “rigorous theoretical analysis” wording, clarify the claim as an empirical analysis with a restricted conditional correctness statement, and make the key expressivity/scaling limitation of the Discrete Transformer explicit. They also point to systematic capacity sweeps, provide new ablations on sub-MLP arity, and clarify that the reported attention head classes are not intended as a complete taxonomy, but can be extended with additional router-like or computational head classes when needed. These clarifications substantially improve the framing, scope, and support for the paper’s claims. Based on the rebuttal, I consider my concerns sufficiently resolved and am increasing my overall recommendation from **4 (Weak Accept)** to **5 (Accept)**.

---

### Official Review · Reviewer_3afQ · 2026-03-12

**Soundness:** 2
**Presentation:** 3
**Significance:** 3
**Originality:** 3
**Overall Recommendation:** 4
**Confidence:** 3

**Summary:**

This work proposes a new architecture "Discrete Transformer" to solve the algorithm extraction problem: discover potentially novel programs directly from models trained on the corresponding algorithmic tasks. The architecture consists of Numerical Attention, Numerical MLP, and linear output head. The key idea is to use differentiable sampling mechanisms to gradually enforce discreteness throughout training. The proposed method has been evaluated on 13 tasks across 3 different algorithmic categories.

**Compliance With Llm Reviewing Policy:**

Affirmed.

**Final Justification:**

I appreciate the authors for further clarifying the limitations of the proposed method in the response. I will maintain my positive score.

**Key Questions For Authors:**

My main concerns are on the empirical evaluation. See weaknesses for details.

**Limitations:**

Yes.

**Strengths And Weaknesses:**

**Strengths**
- This work aims to solve an important problem: extract potentially novel algorithm from neural networks trained on algorithm tasks.
- The proposed method has been compared against MIPS on 13 tasks over 3 algorithm categories, where the proposed method is on par with MIPS over most tasks and show some advantage over the Continuous Dynamic category.
- The paper structure is generally clear and easy to follow.

**Weaknesses**
- Lack of evaluation on novel algorithm discovery: Since part of the appeal of algorithm extraction is to identify novel solutions that potentially learned by the neural networks, there should be some empirical evidence justifying this claim. For example, it would be interesting to test on complex tasks where multiple solutions exist. Related to the novelty aspect, since the hypothesis space of neural networks is architecture dependent, how would using discrete transformer affect the potential solutions that can be learned by the model compared with using normal Transformers or using discrete Transformers but without the sparsity constraints?
- The accuracy of the proposed algorithm is only marginally better than MIPS. On 2 out of 3 tasks where MIPS failed to achieve perfect accuracy, the proposed method also achieves only low accuracy. The only advantage seems to be the exponential_moving_average task. It would be great if the authors can show more tasks of this nature, otherwise, claiming it as "clear advantage" seems a bit of an overstatement.
- The work is largely motivated by the feature superposition hypothesis, in particular, the authors claim that "entangled features encoded in overlapping directions obstruct the recovery of symbolic expressions". This claim has not been empirically validated in the paper. In fact, from the mechanistic interpretability literature, localization methods can often identify high-level symbolic algorithms, where variables are represented in non-axis aligned linear subspaces, e.g., [Conmy et al. 2023](https://openreview.net/forum?id=89ia77nZ8u), [Wu et al. 2023](https://openreview.net/forum?id=nRfClnMhVX&noteId=gEAZY4ftUG). Given this is main motivation of the proposed method, it would be great if the authors can justify this claim with empirical evidence.

---

> ### Author Rebuttal · Authors · 2026-03-31
>
> Thanks for your appreciation and detailed feedback. We are happy to address each of your valuable comments.
>
> W1: *Lack of evaluation ...*
>
> We agree that our wording around "novel algorithm discovery" should be more precise. Our evidence does not support the strongest claim of discovering genuinely human-unknown algorithms. Instead, the paper supports two narrower claims: (i) executable algorithms can be recovered directly from trained model weights without human-written target programs, and (ii) different inductive biases can yield different but mathematically equivalent solutions to the same task. The latter is already supported by Sec. 6.2 / Fig. 7: in *maximum_prev2 / minimum_prev2*, removing sub-MLPs shifts the solution from an MLP-based ReLU arithmetic mechanism to a pure-attention windowed-extrema mechanism; in *spring*, masking immediate history induces an alternative higher-order recurrence equivalent to the standard solution. We will revise the paper to emphasize this evidence and replace "novel algorithms" with "alternative/equivalent solutions recovered de novo from weights" where appropriate.
> We also agree that the hypothesis space is architecture-dependent, and we do not claim that the Discrete Transformer preserves the same solution space as a standard Transformer. Instead, our design biases learning toward a discrete, modular, and symbolically identifiable subset of solutions that can be extracted as executable programs. Compared with standard Transformers, whose representations may be dense and entangled, our scalar residual stream and the function separation between attention and MLP encourage a more disentangled computational history amenable to symbolic recovery. Sparsity further sharpens this bias by reducing soft or redundant mixtures and favoring concise, deterministic computation paths. Thus, our point is not that sparsity necessarily improves task solvability, but that it improves identifiability and extractability of the learned mechanism. Importantly, these constraints do not force a single canonical solution: Sec. 6.2 already shows multiple valid mechanisms within our architecture. We will clarify this distinction between expressivity and extractability in the revision.
>
> W2: *The accuracy ...*
>
> We agree that the phrase "clear advantage" was too strong if interpreted as uniform accuracy gains over MIPS on the shared discrete benchmark. This is not our main empirical message. A more accurate summary is that, on the shared benchmark, our method is broadly comparable to MIPS, while its main qualitative advantage is broader applicability: it supports algorithm extraction in settings with continuous-valued intermediate computations, beyond the discrete tasks in the shared comparison.
> To further support this point, we expanded the evaluation from two to five continuous-valued tasks by adding Linear Drop, Quadratic Drop, and Damped Harmonic Oscillator. These benchmarks cover distinct forms of continuous computation, including real-valued smoothing/aggregation, first-order state updates, and second-order damped oscillatory dynamics. Across three random seeds, the recovered programs achieve average RMSEs of 5.21e-4, 9.37e-4, and 3.18e-4 on these three tasks, respectively. While not exhaustive, these results strengthen the evidence that the framework can recover algorithms with continuous-valued latent computations, rather than only discrete-valued rules. In the revision, we will therefore replace "clear advantage" with the more precise claim: comparable performance on the shared benchmark, with a qualitative advantage in broader applicability to continuous-valued settings.
>
> W3: *The work ...*
>
> We agree that our wording should not be read as claiming that superposition makes mechanistic interpretability impossible in general, since prior work localizes high-level symbolic algorithms even in non-axis-aligned subspaces. Our intended claim is narrower: such entangled representations substantially complicate direct symbolic program extraction in our setting, which requires explicit operands, deterministic routing, and low-dimensional module-wise symbolic regression. To test this directly, we add a post-hoc rotation control: on a trained *sum_last2* model, we apply random orthogonal basis changes to the residual stream and compensate all downstream readout/output weights so as to preserve the model’s predictive accuracy. Under this function-preserving change of basis, blind extraction success drops from 3/3 seeds to 0/3. By contrast, when the inverse rotation is provided as an oracle, recovery returns to near-baseline performance (average RMSE 8.08e-5). This shows that entanglement does not remove information, but makes the explicit operands needed for direct symbolic recovery much less accessible. In the revision, we will replace "obstruct recovery" with "substantially complicate direct symbolic-regression-based recovery" and better distinguish our goal from broader mechanistic-interpretability.

---

> > ### Author Rebuttal · Reviewer_3afQ · 2026-04-02
> >
> > I appreciate the authors for further clarifying the limitations of the proposed method in the response. I will maintain my positive score.

---

### Official Review · Reviewer_PVXC · 2026-03-14

**Soundness:** 3
**Presentation:** 4
**Significance:** 3
**Originality:** 3
**Overall Recommendation:** 4
**Confidence:** 4

**Summary:**

This paper proposes a Transformer architecture, called Discrete Transformer, that is amenable to algorithmic extraction.  The authors replace the residual stream with a collection of scalars, augment attention to operate over scalars, and augment the MLP blocks to operate as a collection of submodules operating over a small number of variables in the residual stream.  The authors show their technique maintains parity with RNN-based prior work and generalizes to continuous settings.

**Compliance With Llm Reviewing Policy:**

Affirmed.

**Final Justification:**

My primary hesitations with this work were a lack of understanding of the relation between the authors' work and Transformer Programs, which I now believe is sufficiently different, and a lack of evaluation on their improved capability for representing continuous computation, which is addressed by the addition of new benchmarks.

**Key Questions For Authors:**

- How well does work like Transformer Programs or Adaptive Transformer Programs perform on the benchmarks you evaluate on?  Or, if these are not relevant baselines, I would like to know why.  The main threat I see to the significance of your paper is whether it is necessary to alter the Transformer architecture to the degree you have, if the goal is algorithmic extraction.  If this question is adequately answered, I am happy to raise my score.

**Limitations:**

The authors adequately address the limitations.

**Strengths And Weaknesses:**

**Strengths**

- The paper is quite well-written.
- Their technique maintains parity with MIPS on existing tasks and demonstrates some generality over MIPS.

**Weaknesses**

- Significance is potentially limited, barring resolution to my question below.
- Given that support for continuous variables is one of the proposed value-adds over prior work, I would like to see more benchmarks demonstrating this capability.
- On L021 (right column), the authors claim the continuous nature of Transformers is problematic for MIPS, but RNNs are also continuous, and they cited the success of the technique for RNNs in the previous paragraph.
- L244: “This prunes unnecessary variables from the computation graph, retaining the
active variables contributing to the final prediction.” This claim is not validated.
- Not sure whether there’s something I’m missing, but Figure 5 is counterintuitive to read, because it seems one is supposed to read it from right to left.
- The claims in Figure 5 are too strong.  The claimed trend for loss and discrepancy only prominently holds for FB and Spring.

---

> ### Author Rebuttal · Authors · 2026-03-28
>
> Thanks for your detailed review. We are happy to address your comments below in the order raised.
>
> W1: *Given that ...*
>
> We agree that broader empirical evidence would strengthen this contribution. To address this, we have expanded the evaluation from two to five continuous-valued tasks by adding Linear Drop, Quadratic Drop, and Damped Harmonic Oscillator. These additional benchmarks were chosen to cover distinct forms of continuous computation: real-valued smoothing/aggregation, first-order continuous state updates, and second-order damped oscillatory dynamics. On these tasks, the extracted programs achieve average RMSEs of 5.21e-4, 9.37e-4, and 3.18e-4, respectively, across three random seeds. While still not exhaustive, these additional tasks substantially broaden the empirical evidence for our claim that the framework can recover algorithms with continuous-valued intermediate computations, rather than only discrete-valued rules.
>
> W2: *On L021 ...*
>
> Thank you for pointing this out. We agree that our wording was imprecise. Our point is not that continuity itself makes MIPS inapplicable, since RNNs are also continuous. Rather, the limitation lies in the MIPS extraction pipeline, which uses post-hoc integer autoencoders to map continuous hidden states into integer tuples, thereby imposing a finite-state approximation. This approximation is less well suited to tasks involving genuinely continuous-valued variables and intermediate computations. We will revise the manuscript to make this distinction explicit.
>
> W3: *L244: ...*
>
> We agree our wording is too strong. In the revision, we will replace it with a more precise statement: pruning heuristically removes variables that are not needed by the reconstructed dependency graph, while preserving predictive fidelity in our experiments. To support this revised claim empirically, we conducted an ablation comparing extracted programs with and without pruning (values in parentheses denote results without pruning):
>
> Sum2 / Diff2 / Par2
>
> RMSE: 1.05e-4 (1.05e-4) / 0 (2.00e-8) / 1.49e-8 (1.49e-8)
>
> Program Length: 28 (34) / 22 (33) / 34 (34)
>
> These results indicate that pruning reduces program length while maintaining predictive fidelity on these tasks.
>
> W4: *Not sure ...*
>
> Thank you for this comment. We will redesign Figure 5 in the revision to make the training trajectory easier to follow and more visually natural.
>
> W5: *The claims ...*
>
> Thank you — we agree. The lag between loss and discrepancy is task-dependent and visually pronounced mainly on more challenging tasks such as FB and Spring. For simpler tasks such as Sum2, Diff2, and Par2, discrepancy tracks the loss more closely, making the lag less apparent. We will revise the text and figure captions accordingly.
>
> Thanks for your key questions, and we will address them in two parts: (1) whether Transformer Programs / Adaptive Transformer Programs are comparable baselines, and (2) whether our architectural changes are necessary for algorithm extraction.
>
> Q1:
> We agree that TP / ATP are among the closest related approaches, and we will clarify this relationship more carefully in the revision. The key distinction lies in both objective and representational scope. These works aim to decompile a transformer’s forward computation into a bounded symbolic formalism, whereas our goal is to recover compact executable algebraic programs via symbolic regression. This distinction matters particularly for continuous-dynamics tasks in our benchmark (e.g., EMA), where real-valued intermediate variables are essential, but it also matters for the discrete tasks, since our target remains algebraic program recovery rather than bounded-form forward decompilation. In other words, our pipeline is designed to extract an interpretable algorithm from the trained model rather than to reproduce the model’s forward pass itself. For this reason, we view TP / ATP as closely related but not fully aligned with the evaluation target in our setting. We will revise the discussion to make these differences in goals, assumptions, and outputs explicit.
>
> Q2:
> We do not claim that such modifications are necessary for task solving in general, nor that standard Transformers are impossible to analyze. Our claim is narrower: these architectural constraints serve as inductive biases that make automated program extraction substantially more tractable in our framework. By exposing scalar variables, encouraging discrete dependencies, and separating attention-as-routing from MLP-as-arithmetic, the architecture reduces the ambiguity that otherwise makes variables, dependencies, and module roles difficult to identify. We will revise the manuscript to make this framing more precise and avoid overstating a formal necessity claim.
>
> We greatly appreciate your valuable suggestions for improving the draft. We hope these revisions address your concerns and clarify the scope and contributions of the paper. If any concerns remain, please do not hesitate to share further feedback.

---

> > ### Author Rebuttal · Reviewer_PVXC · 2026-04-03
> >
> > Thank you for addressing my concerns.  I am satisfied and raise my score to weak accept.

---

### Decision · Program_Chairs · 2026-04-30

**Decision:**

Accept (regular)

**Comment:**

The paper addresses an exciting problem: extracting new algorithms from trained models. I personally find this topic fascinating. I believe the community has something to learn from this paper. All reviewers are satisfied with the paper, so I recommend acceptance.